# Serpentine alteration as source of high dissolved silicon and elevated $\delta^{30}$Si values to the marine Si cycle

Sonja Geilert [1✉], Patricia Grasse [1,2], Klaus Wallmann [1], Volker Liebetrau [1] & Catriona D. Menzies [3]

Serpentine alteration is recognized as an important process for element cycling, however, related silicon fluxes are unknown. Pore fluids from serpentinite seamounts sampled in the Mariana forearc region during IODP Expedition 366 were investigated for their Si, B, and Sr isotope signatures ($\delta^{30}$Si, $\delta^{11}$B, and $^{87}$Sr/$^{86}$Sr, respectively) to study serpentinization in the mantle wedge and shallow serpentine alteration to authigenic clays by seawater. While serpentinization in the mantle wedge caused no significant Si isotope fractionation, implying closed system conditions, serpentine alteration by seawater led to the formation of authigenic phyllosilicates, causing the highest natural fluid $\delta^{30}$Si values measured to date (up to $+5.2 \pm 0.2‰$). Here we show that seafloor alteration of serpentinites is a source of Si to the ocean with extremely high fluid $\delta^{30}$Si values, which can explain anomalies in the marine Si budget like in the Cascadia Basin and which has to be considered in future investigations of the global marine Si cycle.

[1] GEOMAR Helmholtz Centre for Ocean Research Kiel, Wischhofstr. 1-3, 24148 Kiel, Germany. [2] German Centre for Integrative Biodiversity Research (iDiv) Halle-Jena-Leipzig, Deutscher Platz 5e, 04103 Leipzig, Germany. [3] Department of Earth Sciences, Durham University, Science Laboratories, South Road, Durham, UK. ✉email: sgeilert@geomar.de

Serpentinites are expected to play a fundamental role in the exchange of Si between the Earth's mantle and the global ocean since dissolved Si is taken up during serpentine formation[1]. Serpentine is a hydrated Mg-silicate $(Mg_6Si_4O_{10}(OH)_8)$ which forms during the reaction of hydrous fluids with mantle rocks and occurs in a variety of marine settings including slow-spreading mid-ocean ridges, rifted continental margins, and fore-arc regions of subduction zones under a wide range of temperature and pressure regimes[2–6]. At slow and ultra-slow spreading ridges, serpentinites can make up to 20% of the sea-floor, extending 3–4 km into the footwall as seawater circulates through fractures and faults[6,7]. Despite the increasing awareness of the importance of serpentinization influencing global element cycles, serpentine alteration (here defined as weathering reaction after preceding serpentinization) during water-rock interactions at low temperatures (<20 °C) is not well understood. During alteration reactions, Si is removed from mafic or ultramafic rocks and partially re-precipitates as authigenic clay minerals, nevertheless resulting in a net gain of Si to the ocean[8]. However, this Si flux is associated with an unknown Si isotope composition ($\delta^{30}Si$) adding a large uncertainty to models simulating the global Si cycles. Dissolution and precipitation processes have been found to induce significant Si isotope fractionation during biotic and abiotic processes (see recent reviews by Frings et al.[9] and Sutton et al.[10]), spanning a natural range in solid phase $\delta^{30}Si$ values between −5.7‰ (silcretes)[11] and +6.1‰ (phytoliths)[12] and in fluid $\delta^{30}Si$ values between −2.05‰ (soil solutions)[13] and +4.66‰ (freshwater)[14]. Pacific Ocean $\delta^{30}Si$ values are mainly controlled by diatom uptake in surface waters, subsequent dissolution and water mass mixing, spanning a range from +0.8‰ in Northern Pacific deep water masses to +4.4‰ in the photic zone (average deep Pacific: $+1.2 \pm 0.2$‰, 1 standard deviation (SD)[15,16]; for a compilation see Sutton et al.[10] and Grasse et al.[17]). During IODP Expedition 366, pore fluids from serpentinite seamounts in the Mariana and Izu-Bonin arc region were sampled, in order to study Si isotope fractionation during serpentinite-seawater alteration reactions. In addition, radiogenic Sr ($^{87}Sr/^{86}Sr$) and stable B isotopes ($\delta^{11}B$) were investigated to further unravel fluid sources and fractionation mechanisms. Serpentine can incorporate large amounts of B in its crystal structure (up to $100 \mu g\,g^{-1}$) and preferentially incorporates the $^{10}B$ isotope, enriching associated fluids with the $^{11}B$ isotope[18–21]. Despite this distinct fractionation behavior, a large range of $\delta^{11}B$ values in serpentinites has been detected to date, ranging from −15.3 to +40.7‰[18–23]. B isotopes have been used to study mineral reactions as B is a fluid mobile element and high amounts of B can be incorporated in the serpentine crystal lattice, when B concentrations in the fluids are high ($D_{(serp/H2O)} = 0.25$[24]). Therefore, B is a useful tracer to identify fluid-rock processes[25], including serpentine alteration. Coupling of Si and B isotopes should result in a positive correlation, in that the light isotope is incorporated in the authigenic mineral for both isotope systems. The combination of a new tracer (Si) with a well-established tracer (B) will help to reveal processes during seawater alteration of serpentine and authigenic mineral precipitation. The isotopic results were further evaluated by transport-reaction modeling to quantify rates of serpentine alteration reactions in contact with seawater and isotopic fractionation.

Our findings show that serpentine dissolution and the subsequent formation of authigenic minerals is a source of Si and B to the ocean with very high $\delta^{30}Si$ and $\delta^{11}B$ values. We hypothesize that the results from serpentine alteration are directly transferable to basalt alteration on the seafloor due to similar geochemistry of the authigenic mineral assemblages. With regard to Si cycling, seafloor alteration reactions result in the release of isotopically heavy Si, which likely impacts marine water mass isotope signatures and may explain local anomalies in the marine Si cycle, for example in the Cascadia Basin.

## Results and discussion

The Mariana and Izu-Bonin forearc is the only known region on Earth where serpentinite seamounts form above a non-accretionary convergent plate margin. Permeable faults serve as long-lived pathways for deep-sourced fluids and serpentinized mud to ascend to the surface[26–28]. Two seamounts were investigated (Yinazao and Fantangisña), which are located at 55 km and 62 km from the trench axis, respectively (Fig. 1a, b). Three drill cores were recovered from Yinazao (1491B and C, 1492B) and three from Fantangisña seamount (1498A and B, 1497B; Fig. 1a, b; Supplementary Table 1). Drill cores were taken from the flanks of the seamounts (except coring location 1497B, which was located close to the seamount summit), in order to study mineral reactions during water-rock reactions induced by seawater circulating through the mount flanks. Thermal gradients induce temperature changes of ≤2 °C from the shallowest (about 4 °C) to the deepest sample (about 6 °C) for both seamounts[28] and reaction temperatures are thus similar between the individual samples with regard to the maximum seawater penetration depth of about 30 m. Recovered material in the uppermost 4–5 m core sections consisted of pelagic clays at Yinazao and sandy silt at Fantangisña seamount. Deeper parts of the seamounts were composed of serpentinite mud which contained xenoliths of the underlying forearc crust and mantle, and the subducting plate[28]. A detailed description of the lithostratigraphic units of the investigated seamounts can be found in Fryer et al.[28].

The Si concentrations in fluids from both seamounts vary unsystematically with depth (range from 44 to 516 µM Si, Supplementary Table 1; NW Pacific seawater: 146 µM Si[15]). In contrast to the Si concentrations, Si isotope values generally increase from +0.4‰ to +5.2‰ with depth, with the majority being higher compared to deep NW Pacific seawater ($\delta^{30}Si_{SW}$: +1.05‰[15]). The maximum $\delta^{30}Si$ value in the fluids sampled at the mount flanks ($+5.2 \pm 0.2$‰; $2SD_{external\ reproducibility}$) constitutes the highest natural fluid $\delta^{30}Si$ measured to date (Fig. 1c). In contrast to the large isotope variability observed in the fluids, serpentinite muds show only a small range in $\delta^{30}Si$ between −0.6‰ and +0.1‰, independent of seamount and sampling site (Supplementary Table 2, Supplementary Fig. 1). The fluids are further characterized by high B concentrations (from 245 to 758 µM B) with a wide range in $\delta^{11}B$ (+16.1 to +43.5‰), which encompasses $\delta^{11}B$ of seawater (seawater B concentration: 432.6 µM[29]; $\delta^{11}B_{SW}$: +39.6‰[30]) (Fig. 1c).

The combination of pore fluid $\delta^{11}B$ and $\delta^{30}Si$ values shows a very good correlation of these two isotope systems (Fig. 2). For both systems, the isotope values are below and above the seawater signature, indicating dissolution of presumably serpentine and precipitation of authigenic mineral phases, respectively. In order to decipher the processes controlling mineral dissolution and precipitation, fluid sources and compositional changes in relation to depth need to be examined.

**Origin of pore fluids**. Two major fluid sources can be distinguished based on magnesium ($Mg^{2+}$), strontium (Sr), and chloride ($Cl^-$) concentrations, radiogenic Sr isotopes, and pH (Figs. 3, 4a–c).

Most of the pore fluids show seawater-like signatures in the upper ~30 m (surface layer), except samples from Fantangisña sites 1498 A and B (Figs. 3, 4a–c). Samples from site 1498B overlap with the local fluid mantle endmember and can thus be identified as mantle fluids, showing depleted $Mg^{2+}$ concentrations, high pH values (~11), decreasing $Cl^-$ concentrations, and

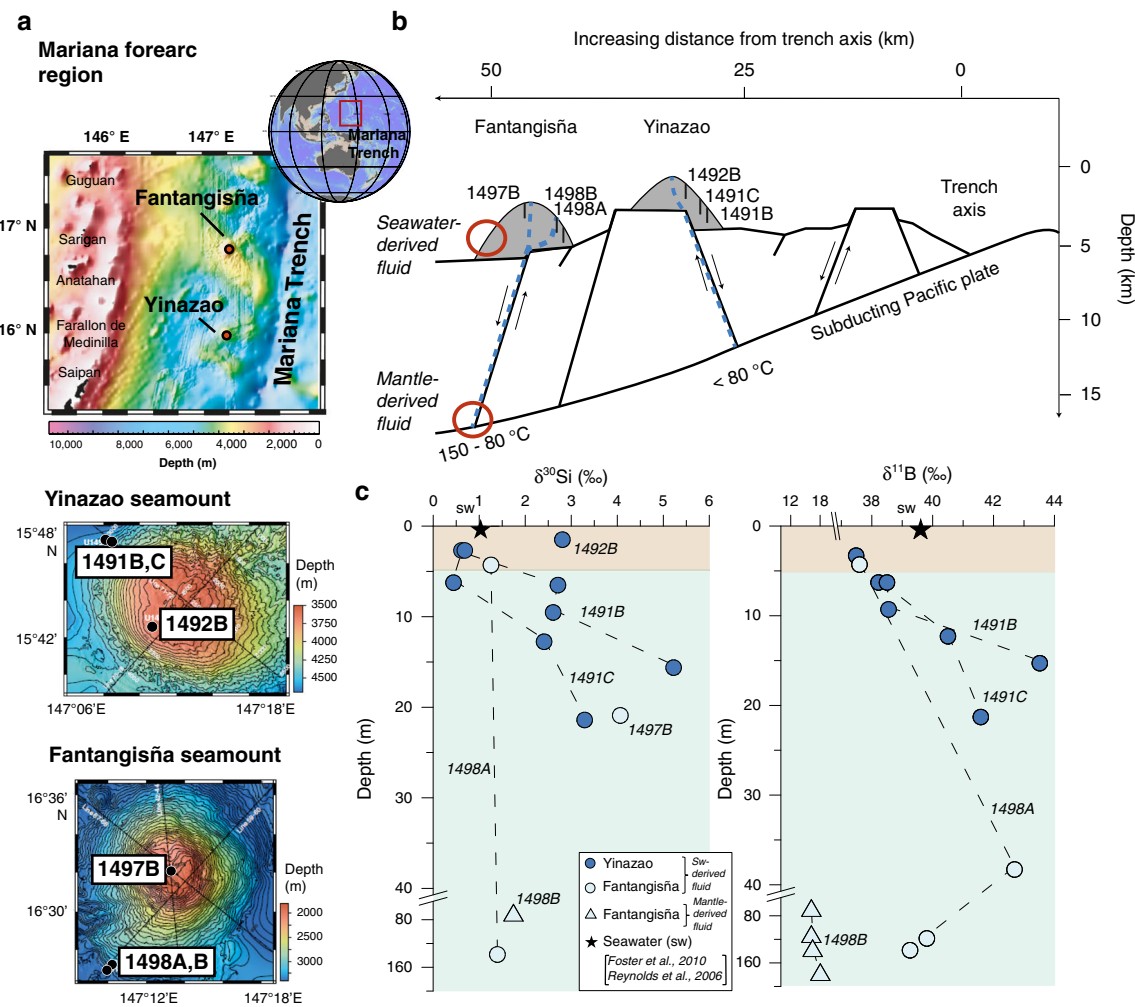

**Fig. 1 Sampling location and isotope results. a** Sampling area of the two seamounts (Yinazao and Fantangisña) investigated during IODP Expedition 366 and close up of the sample locations for both seamounts and (**b**) sketch of the Mariana forearc region with indicated sampling locations (modified after Fryer et al.[28]). Blue dotted lines indicate fluid flow along faults. Note mantle-derived fluid flow for site 1498B (see sections "Origin of pore fluids" and "Drivers of deep mantle-derived $\delta^{30}Si$ and $\delta^{11}B$ values"). **c** Isotope compositions versus depths (m) are displayed for $\delta^{30}Si$ and $\delta^{11}B$. Error bars (2 SD of individual measurements) are within symbol size. Brown area denotes depth of uppermost pelagic clays ($\leq 5$ m depth) and green area shows mineralogy dominated by serpentinites and mafic xenoliths.

increasing Sr concentrations, characteristic trends reported for deep mantle fluids from the Mariana forearc region[26,27]. Also, a $^{87}Sr/^{86}Sr$ ratio similar to deep-sourced mantle fluids is measured ($^{87}Sr/^{86}Sr$ of 0.70535: ODP Site 1200 at South Chamorro Seamount[26]) (see also lower red circle in Fig. 1b). Therefore, we interpret the related Si and B isotope values to originate from the deep mantle affected by pervasive serpentinization. Sample 1498 A 15-R-1 shows a high Sr concentration (189 μM) and a more radiogenic $^{87}Sr/^{86}Sr$ signature (0.70763; Fig. 4) compared to the mantle-derived fluids and likely results from mixing between the two endmembers (seawater and deep-mantle fluid) discussed above.

**Drivers of deep mantle-derived $\delta^{30}Si$ and $\delta^{11}B$ values.** During serpentinization, pore fluid pH can increase rapidly to alkaline values (pH ~ 9–12[28]), so that dissolved B is tetrahedrally coordinated. This is also the species preferentially bound in the serpentine mineral structure, and so limited fractionation between mineral and water is expected taking also the high formation temperatures into account (average $T$ of 200 °C)[20,31]. The mantle-derived pore fluids of site 1498B have the lowest $\delta^{11}B$ values of about +16‰ and the lowest B concentrations of the investigated

fluids (between 84 and 111 μM; Supplementary Table 1). These low $\delta^{11}B$ values overlap within error with serpentinized peridotite clasts and serpentinite matrix identified by Benton et al.[18] with $\delta^{11}B$ values of about +14‰ for Conical Seamount in the Mariana forearc. Consequently, we conclude that no significant B isotope fractionation occurs during early pervasive high pH serpentinization reactions. In contrast, Si isotopes vary between potentially mantle-derived fluids ($\delta^{30}Si = +1.6‰$) and serpentinite muds (on average $\delta^{30}Si = -0.3 \pm 0.2‰$; 1 SD; Supplementary Table 2). However, by comparing pristine mantle rocks ($\delta^{30}Si = -0.29 \pm 0.08‰$)[32] with the serpentinized muds investigated in this study, we show no, or only minor, Si isotope fractionation occurs during the transformation of olivine/ pyroxene to serpentine. This also confirms the isochemical nature of pervasive serpentinization[3]. The pore fluid $\delta^{30}Si$ value is likely affected during ascent of the serpentinite muds and accompanied cooling, which induces Si precipitation and fractionation of Si isotopes.

**Processes of serpentine alteration.** The investigation of Si and B isotopes revealed a similar fractionation response during serpentine alteration and shows the potential of coupled Si and B isotope data to trace serpentine alteration reactions (Fig. 2).

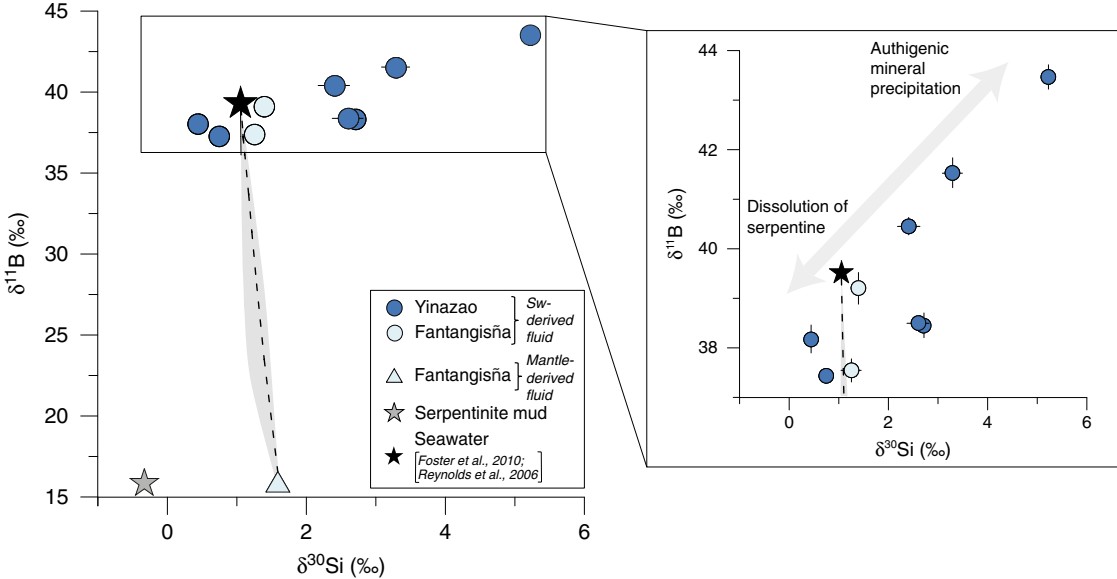

**Fig. 2 Pore fluid $\delta^{11}B$ versus $\delta^{30}Si$ values reveal different stages of silicate dissolution and Si precipitation.** The mixing curve is calculated following Eq. (1) between seawater and a mantle-derived fluid. The gray-shaded area takes the varying low Si concentrations of the mantle-derived fluids into account (see section "Methods", "Mixing curve between seawater and a mantle-derived fluid" for details). Error bars (2 SD of individual measurements) not indicated are within symbol size. Note that the $\delta^{30}Si$ value of the mantle-derived fluid is likely affected by Si precipitation during fluid ascent and that the data point from the serpentinite mud is derived from direct $\delta^{30}Si$ measurements (Supplementary Table 2) and inferred for $\delta^{11}B$ from pore fluids given that no fractionation occurs during serpentinization (detailed in section "Drivers of deep mantle-derived $\delta^{30}Si$ and $\delta^{11}B$ values").

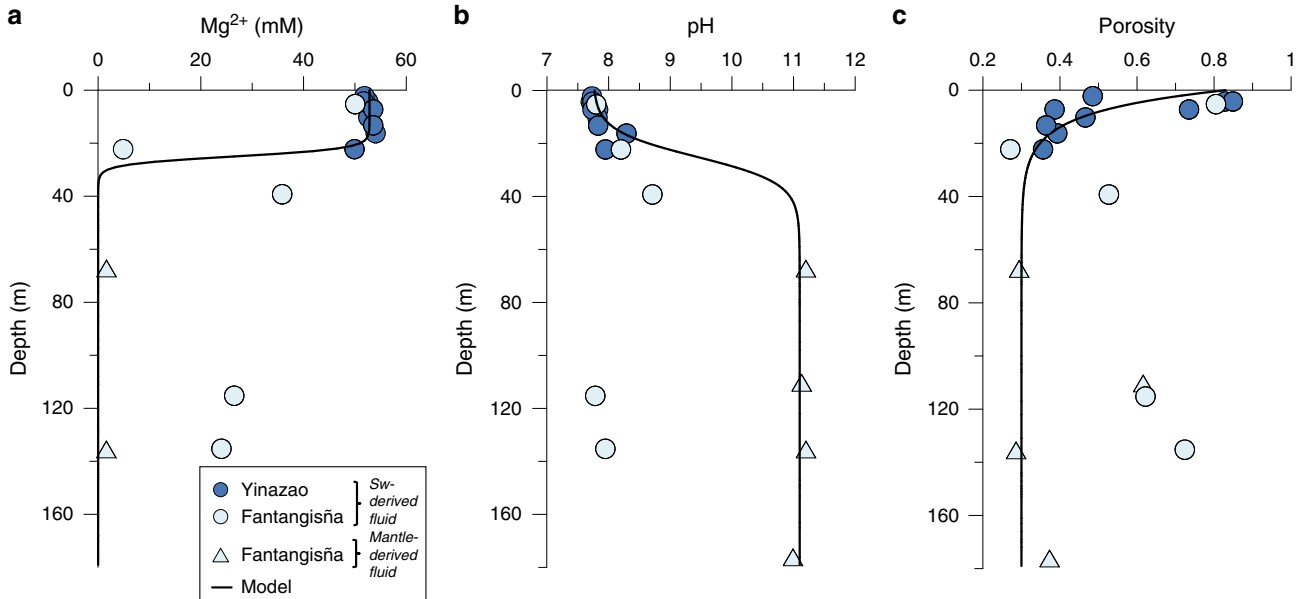

**Fig. 3 Depth-profiles characterizing the layering of the seamount fluid system. a** $Mg^{2+}$ concentrations in pore fluids, (**b**) pH values, (**c**) sediment porosity. Functions were fitted through the data to generate continuous depth profiles for the transport-reaction modeling (Supplementary Methods). $Mg^{2+}$ and porosity data are reported in Fryer et al.[28].

Interestingly, despite generally increasing Si and B concentrations, the isotope values of both elements increase as well. This observation seems counterintuitive, in that concentration increases generally relate to mineral dissolution processes, which are not associated with significant isotopic fractionation[33]. Motivated by this observation, a one-dimensional transport-reaction model (see "Methods" section and Supplementary Methods for model details) was set up to constrain—for the first time—the seawater entrainment rate, the precipitation and dissolution rates, and the isotopic fractionation during authigenic

mineral formation. This model is a simplification as it does not capture the additional changes which are likely induced by the 3-D geometry of the seamounts, and the time-dependency and space-dependency of mixing rates and fluid flow velocities.

The depth profiles indicate a seamount surface layer with a thickness of about 30 m where bottom water is entrained into the muds by shallow seawater circulation and an underlying layer that is affected by the ascent of deep fluids originating from the subducted slab (Figs. 3, 4). The strong increase in Si and B concentrations in the surface layer is induced by rapid dissolution of serpentine, which is

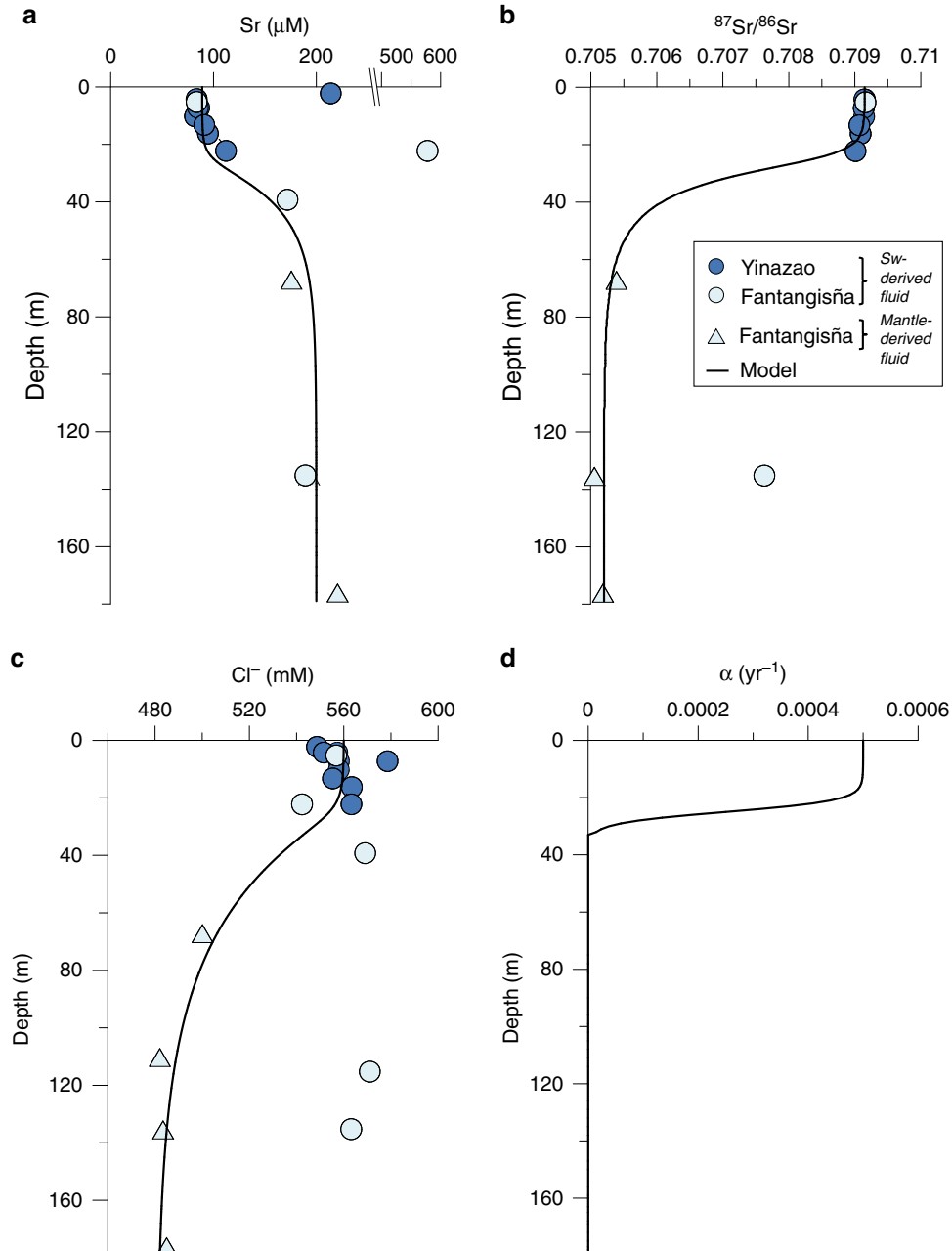

**Fig. 4 Depth profiles of conservative tracers. a** Dissolved Sr, (**b**) $^{87}Sr/^{86}Sr$ ratios in pore fluids, (**c**) Dissolved Cl$^-$ (taken from Fryer et al.[28]), (**d**) The non-local mixing coefficient α was derived by fitting the model to the data (**a**–**c**). Dissolved Cl$^-$, Sr, and $^{87}Sr/^{86}Sr$ ratio in the pore fluid samples were employed to constrain the steady-state mixing rate and fluid flow velocity assuming that the depth-distribution of these dissolved tracers is governed by transport processes rather than precipitation/dissolution reactions. The best fit to these data was obtained applying mixing in the upper 30 m with a non-locale mixing coefficient of $5 \times 10^{-4}\,yr^{-1}$ and an upward fluid flow velocity of 0.01 cm yr$^{-1}$. Thick lines indicate steady-state model results (Supplementary Methods).

undersaturated due to the entrainment of almost pH-neutral seawater (Fig. 5). The persistent mixing in the surface layer with seawater supports high dissolution rates (~1.2 μmol cm$^{-3}$ yr$^{-1}$) whereas the rates are low in the deeper layers (<10$^{-5}$ μmol cm$^{-3}$ yr$^{-1}$) that are not affected by mixing with ambient seawater (Fig. 5). Serpentine, which formed in the mantle wedge, becomes unstable after deposition on the seafloor either as serpentinized clasts or as serpentinite mud and begins to dissolve during interaction with seawater via hydrolysis[34], following:

$$Mg_3(Si_2O_5)(OH)_4 + 5H_2O \rightarrow 3Mg^{2+} + 2H_4SiO_4 + 6OH^-,$$

thereby releasing Si and B into the pore fluid (see also upper red

circle in Fig. 1b; Fig. 6a, b). The δ$^{30}$Si and δ$^{11}$B values increase in the about upper 30 m of the seamounts even though isotopically depleted Si (δ$^{30}$Si = −0.3‰; Supplementary Table 2) and B (δ$^{11}$B = +16‰; see section "Drivers of deep mantle-derived δ$^{30}$Si and δ$^{11}$B values") are released by the dissolving serpentine. This observation can only be explained by formation of authigenic minerals removing isotopically depleted Si and B from the pore fluids (Figs. 2, 6). The formation of authigenic minerals was simulated by allowing talc (Mg$_3$Si$_4$O$_{10}$(OH)$_2$) precipitation in the model. Talc was chosen because thermodynamic equilibrium calculations conducted with PHREEQC[35] show that the surface layer of the seamounts is strongly oversaturated with respect to

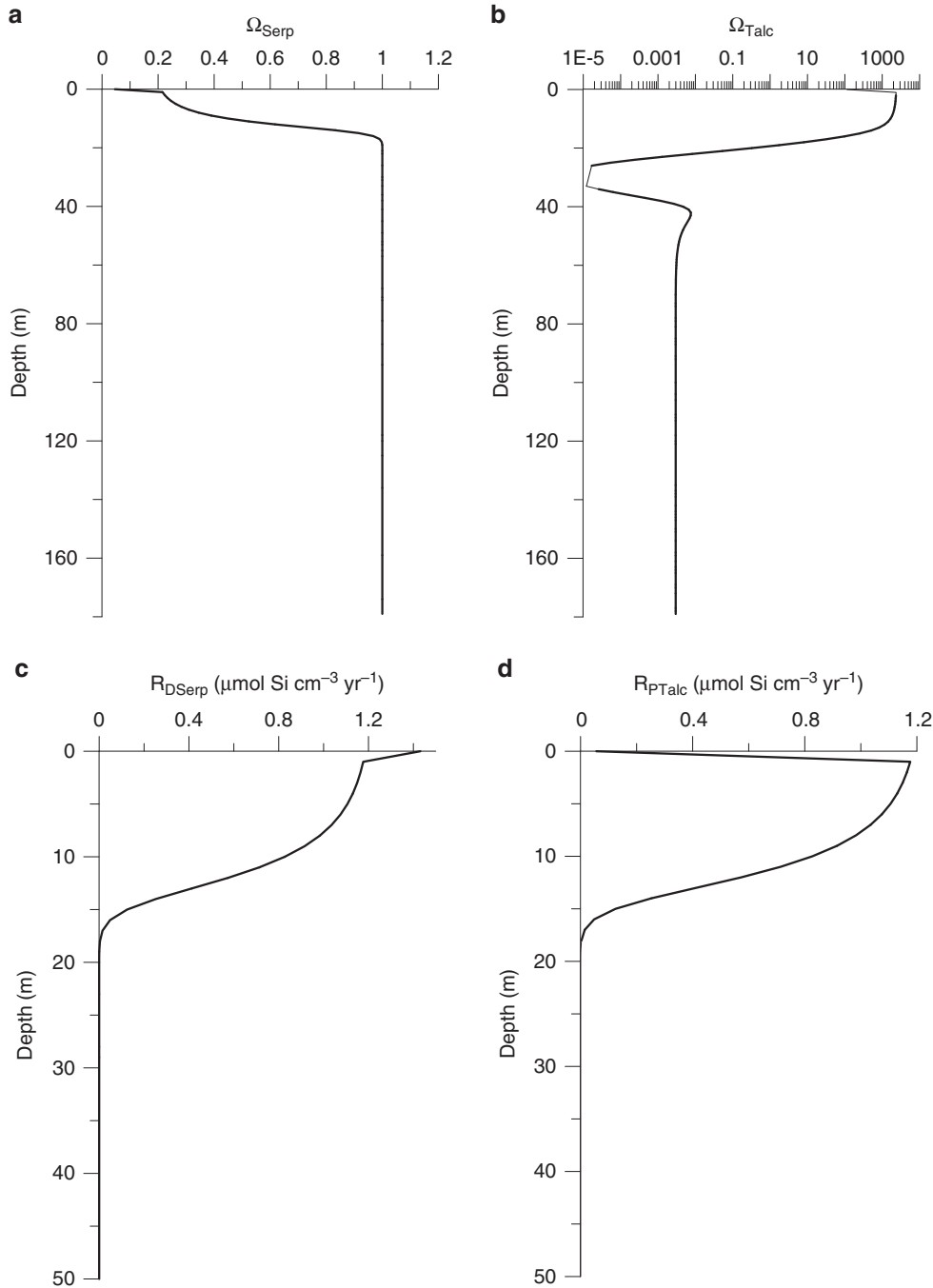

**Fig. 5 Reaction rates applied in the modeling. a** Saturation state of the pore fluid with respect to serpentine, (**b**) saturation state of the pore fluid with respect to talc. Saturation states were calculated applying measured pH values and $Mg^{2+}$ concentrations (Fig. 3) and modeled silica concentrations (Fig. 6). **c** Rate of serpentine dissolution, (**d**) rate of talc precipitation. Note the different depth scale in **c** and **d**.

this mineral (Fig. 5). Other Al-containing silicate phases may form as well; however, we were not able to simulate the formation of these phases due to the lack of dissolved Al data. The modeling indicates that a large fraction of the Si and B released from the dissolving serpentine is removed from solution by authigenic mineral precipitation (Fig. 5). We obtained a good fit to the $\delta^{30}Si$ data applying a Si isotope fractionation of $\Delta^{30}Si = -3‰$, where $\Delta^{30}Si$ is defined as $\Delta^{30}Si = \delta^{30}Si_{mineral} - \delta^{30}Si_{pore\ fluid}$. Lower and higher Si isotope fractionation could not reproduce the observed $\delta^{30}Si$ values of the sample as tested by model runs using $\Delta^{30}Si$ of $-2‰$ and $-4‰$ (Fig. 6c). This Si isotope fractionation is the highest abiotic fractionation observed for authigenic mineral

formation so far. In natural systems, only Si precipitation from supersaturated solutions to form amorphous silica and experimentally investigated Si adsorption to Al shows $\Delta^{30}Si$ values in this range or higher (up to $-5‰$)[36–38].

The almost complete removal of dissolved Si by authigenic mineral precipitation (~99% Si removed from pore fluid) induces a strong $\delta^{30}Si$ maximum at the base of the surface layer, which is the highest pore fluid $\delta^{30}Si$ value that has been observed in natural environments. There is only a limited data set available on marine pore fluid $\delta^{30}Si$, mostly originating from continental margin settings. The $\delta^{30}Si$ values range between $-0.5$ to $+2.5‰$ and are dominantly controlled by dissolution of biogenic silica

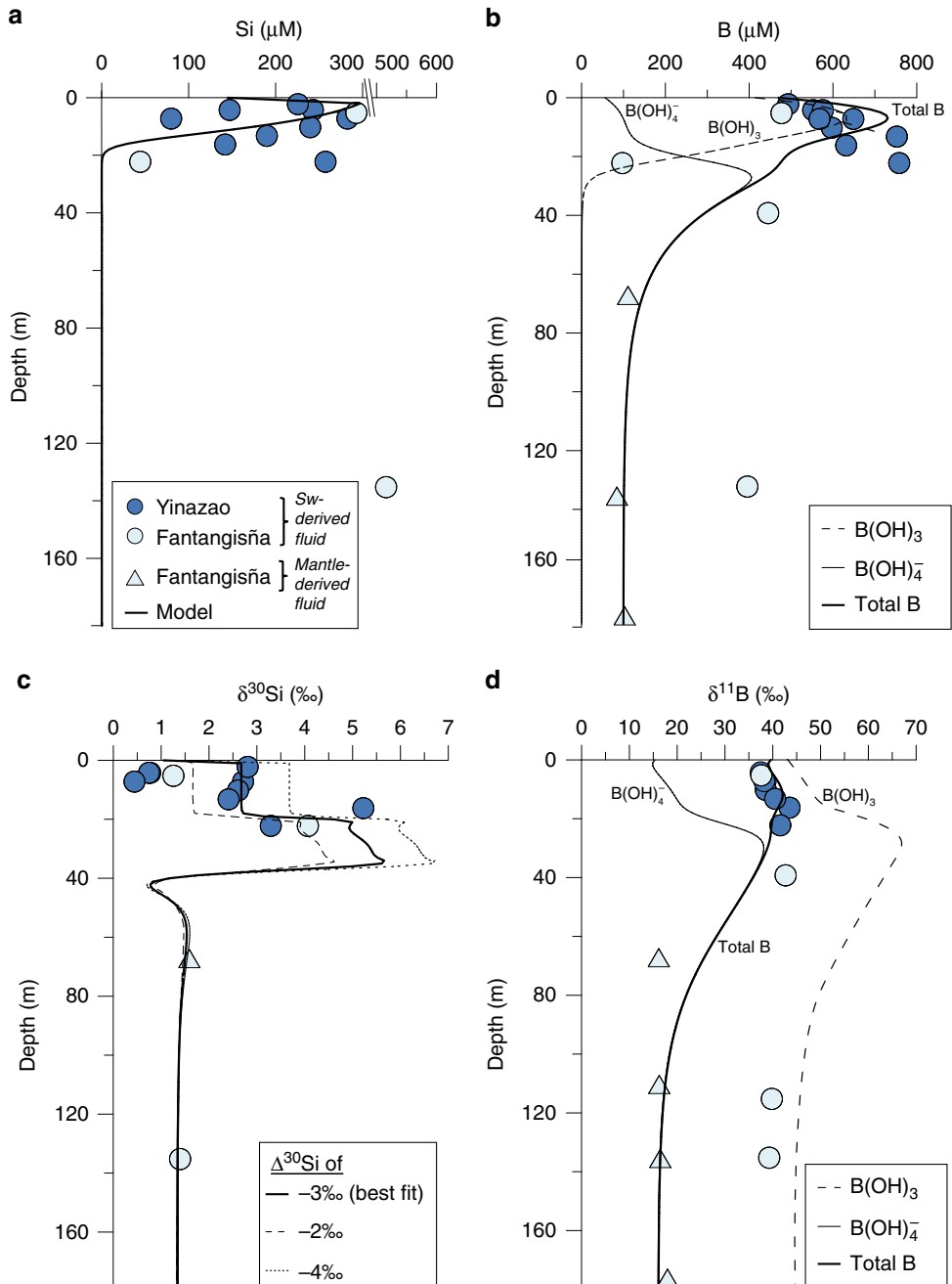

**Fig. 6 Depth profiles of reactive tracers. a** Dissolved Si, (**b**) total dissolved B, dissolved borate (B(OH)$_4^-$) and dissolved boric acid (B(OH)$_3$), (**c**) $\delta^{30}$Si of dissolved Si. Model data based on a Si isotope fractionation with $\Delta^{30}$Si of $-3$‰ (best fit) and sensitivity tests assuming $-2$‰ and $-4$‰ (dotted lines). **d** $\delta^{11}$B values of total dissolved B, dissolved borate and dissolved boric acid following Zeebe and Wolf-Gladrow[45] and Klochko et al.[46] (see section "Processes of serpentine alteration" and Supplementary Methods for details). Error bars (2 SD of individual measurements) within symbol size. Thick or dashed lines indicate steady-state model results (Supplementary Methods).

and the formation of authigenic clays with a $\Delta^{30}$Si of $-2$‰[39–41]. The Si isotope fractionation depends on the chemical composition and crystal structure of the authigenic mineral, the precipitation rate, and temperature[36,37,42,43]. We can only speculate that a combination of slow reaction rates associated with low temperatures (authigenic mineral precipitation rate ~1.2 μmol Si cm$^{-3}$ yr$^{-1}$ at ~4 °C; Fig. 5) and a Mg-rich authigenic mineral composition drives the Si isotope fractionation to higher values at the Mariana seamounts compared to continental margin settings (e.g., Peruvian margin) with faster reaction rates at higher temperatures (authigenic mineral precipitation rate up to ~27

μmol Si cm$^{-3}$ yr$^{-1}$ at ~11 °C; Ehlert et al.[39]) and an Al-rich authigenic mineral composition. This assumption needs to be tested in further studies, however, similar reactions and associated Si isotope fractionation as observed for the Mariana seamounts may occur in other settings, where oceanic crust and serpentinite interact with seawater at low temperatures (see section "Impact of seafloor alteration on the global Si (isotope) cycle").

The isotopic fractionation of B during mineral precipitation is related to two processes: the pH-dependent fractionation between borate and boric acid in the pore fluid and the fractionation during borate uptake in the solid phase[31]. Hence, we calculated the

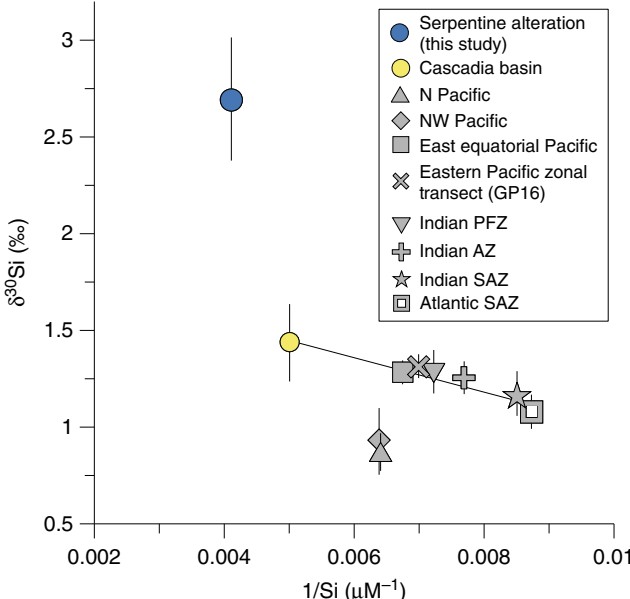

**Fig. 7 Mariana seamount δ³⁰Si value in the global context.** Mariana seamounts median fluid $\delta^{30}Si$ value (error bar equals the coefficient of quartile deviation) versus the inverse Si concentration ($1/Si$ in $\mu M^{-1}$). In addition, Pacific seawater, Subantarctic Zones (SAZs), Indian Antarctic Zone (AZ), Indian Polar Frontal Zone (PFZ), and the Cascadia Basin are shown for waters below 2000 m (modified after Beucher et al.[58], Hendry and Brzezinski[59], and Grasse et al.[17] and references therein). Note that most seawater data fall on a single mixing line (shown in black), between waters of the Cascadia Basin and the SAZs. High Si concentrations and $\delta^{30}Si$ values in the Cascadia Basin might result from serpentine alteration (see text for details).

concentrations of borate and boric acid and their isotopic composition applying the dissociation constant for boric acid in seawater[44] and the isotopic equilibrium constant for borate and boric acid[45]. The model results confirm that dissolved borate has a much lower $\delta^{11}B$ than total dissolved boron in the surface layer (Fig. 6d) such that $^{10}B$ is removed from solution when borate is preferentially bound in the authigenic silicate phase. The modeling showed that an additional fractionation between dissolved borate and solid phase borate has to be applied to match the $\delta^{11}B$ data. The best fit was attained applying an isotopic fractionation factor of $\Delta^{11}B_{mineral-borate}$ of $-20‰$ where $\Delta^{11}B_{mineral-borate}$ is defined as $\delta^{11}B_{borate\ authigenic\ mineral}-\delta^{11}B_{dissolved\ borate}$. This $B_{borate}$ isotope fractionation results in an average $\delta^{11}B$ value of the authigenic clay of $-2.4 \pm 2.0‰$ (1 SD), supposing an average of the modeled $\delta^{11}B_{borate}$ in the upper 15 m (highest talc precipitation rate) of $+17.6‰$. This results in a general B isotope fractionation between authigenic mineral and the pore fluid of $-43.3 \pm 2.3‰$ (1 SD), with $\Delta^{11}B_{mineral-pore\ fluid} = \delta^{11}B_{mineral}-\delta^{11}B_{pore\ fluid}$ (average of the modeled $\delta^{11}B_{pore\ fluid}$ in the upper 15 m of $+40.9‰$). This B fractionation fits well to modeled values for low temperature phyllosilicate formation by Boschi et al.[19] based on data by Liu & Tossel[31].

The coupling of Si and B isotopes showed that the light isotope is preferentially incorporated in authigenic minerals and that the combination of both isotope systems can trace serpentine alteration reactions (Figs. 2, 6). Pore fluids with $\delta^{30}Si$ and $\delta^{11}B$ values lower than seawater ($\delta^{30}Si$ from $+0.4‰$ to $+0.8‰$; $\delta^{11}B$ from $+37.5‰$ to $+38.2‰$; Figs. 1, 2) originate from the shallow, uppermost part of the seamounts (depth < 5 m depth; except samples from 1498B, which originate from the mantle, see also section "Origin of pore fluids"). A possible reason for these low isotopic values might be the

dissolution of authigenic minerals, which formed during potentially higher upward fluid flow velocities in the past. Lower fluid flow velocities in the present would cause dissolution of these authigenic phases as seawater is the dominant ambient fluid phase. Fluid flow velocities can vary significantly between the seamounts in the Mariana forearc region (i.e., between $2\ mm\ yr^{-1}$ to $4\ cm\ yr^{-1}$)[26] and thus changes over time are not unlikely as well. More data are needed to unequivocally disentangle additional process occurring during serpentine alteration, however, our data show for the first time the potential of coupled Si and B isotope data to trace serpentine alteration reactions.

**Impact of seafloor alteration on the global Si (isotope) cycle.** The findings of this study have important implications for the global oceanic Si cycle and isotope budget, as seafloor alteration of serpentinites is found to be a source of Si with extremely high $\delta^{30}Si$ values. So far, none of the global marine Si models or marine budget calculations have considered low temperature serpentine alteration, which might be a common process as serpentinites are widespread on the seafloor especially in the vicinity of Mid Ocean Ridges (MOR), transform and bend faults, and subduction zones[5]. In general, alteration of serpentinites by circulating seawater has received little attention, even though previous studies have shown that low temperature weathering of abyssal serpentinites leads to significant modifications in mineralogy, chemical composition, and physical properties[46,47]. Serpentinites may contribute up to 20 vol.% of the oceanic basement close to MORs and up to 16.7 vol.% in the vicinity of subduction zones[6,7]. Serpentine alteration results in the formation of authigenic minerals like aragonite, Fe-oxyhydroxides, and clays, for example in the abyssal serpentinites at the Iberia abyssal plain[34], the Puerto Rico Trench[46], and from the Mid Atlantic Ridge[48]. The clay mineral assemblages are identified as Mg–Fe-smectite, with saponite and montmorillonite as dominating phases[34,49]. Saponites, as well as other authigenic clay minerals are reported as authigenic minerals in Yinazao and Fantangisña seamounts[50] and resemble in their mineralogy (Mg-rich silicates) closely talc, the mineral phase used in the model approach (see section "Processes of serpentine alteration"). Therefore, we propose that geochemical results from the altered Mariana forearc serpentinites can serve as an example for alteration of oceanic serpentinites in general, implying that alteration reactions of marine serpentinites result in elevated Si concentrations and high $\delta^{30}Si$ values of reacting fluids. Further, we hypothesize, that serpentine alteration reactions are directly comparable to low temperature, off-axis basalt alteration, given that the most abundant authigenic mineral replacing igneous phases is also saponite[51]. Effusing fluids associated with off-axis basalt alteration have Si concentrations between 200 and 550 $\mu M$ Si[52,53], which is identical to the range observed in this study. According to our new data the associated Si fluid isotope signature with serpentine and likely basalt alteration may be high (on average $+2.7 \pm 0.3‰$, locally even up to $+5.2‰$; Supplementary Table 1). We further hypothesize, that the Si isotope signature of the oceanic crust shifts away from the basaltic endmember to lower values with increasing open-system alteration states due to increasing formation of authigenic clays. This process might additionally be enhanced by an increasing degree of mineral crystallization with crustal age[51] and an associated lower clay $\delta^{30}Si$ value[54]. To date, the only direct measurement of altered oceanic crust has an average $\delta^{30}Si$ value of $-0.32 \pm 0.06‰$[55], overlapping with pristine mantle rocks ($\delta^{30}Si = -0.29 \pm 0.08‰$)[32]. However, this value of altered oceanic crust originates from the east side of the East Pacific Rise, where the alteration degree is <10%[56] and thus too small to influence bulk rock $\delta^{30}Si$ values. The impact on oceanic

$\delta^{30}Si$ values needs to be assessed by constraining the Si fluxes and associated $\delta^{30}Si$ values related to serpentine and basalt alteration and authigenic mineral formation. Oceanic crust alteration may also influence oceanic $\delta^{30}Si$ values at regional scale, especially in restricted basins and may potentially explain, at least in part, Si anomalies like the Northeast Pacific Silicic Acid Plume (NPSP). The NPSP has unusually high Si concentrations (>150 μM Si) and high $\delta^{30}Si$ signatures of +1.5‰, the highest Pacific deep ocean Si isotope value[57,58] (Fig. 7), between 2000 and 3000 m water depths, originating mostly from the Cascadia Basin (about 200 μM Si)[59]. This has not been explained unambiguously to date. Findings of this study show that low temperature alteration of serpentinites and potentially seafloor basalts are associated with high Si concentration and $\delta^{30}Si$ values which are expelled to the ocean. Such a source is likely to contribute to the unusually high NPSP and related Si isotope values (see Supplementary Discussion for details). Low temperature seafloor may contribute significantly to the $\delta^{30}Si$ budget of the ocean, considering the large areas of serpentinized oceanic crust and the related fluid discharge with extremely high Si isotope values.

## Methods

**Shipboard sampling and pore fluid analyses.** A detailed description of the sampling and analytical procedures on the JOIDES resolution during IODP expedition 366 can be found in Fryer et al.[28]. In short, pore fluids were processed from whole-round core samples immediately after core retrieval in a nitrogen-filled glove bag. The inner part of each core section was further processed to avoid seawater contamination during retrieval and compressed in a titanium squeezer by a hydraulic press. The pore fluids were filtered with a pre-cleaned Whatman No.1 filter in the titanium squeezer and after extraction filtered through a pre-cleaned Gelman polysulfone disposable filter. Pore fluid major element analyses were conducted by ICP-OES on board following Gieskes et al.[60] and IODP standard analytical protocols. The IAPSO seawater standard was used for assessment of the measurement uncertainty and the uncertainty was ~5%. The pH was measured immediately after pore fluid extraction following Gieskes et al.[60] using a glass electrode. Repeated measurements of the IAPSO seawater standard yielded a precision for the pH measurements better than 0.01 pH units. Some Sr, Si, and B concentrations were re-measured by ICP-OES (VARIAN 720-ES) at the GEOMAR Helmholtz Centre for Ocean Research Kiel. All analyses were tested for accuracy and reproducibility using the IAPSO salinity standard[60].

**Silicon sample preparation and isotope analyses by MC-ICPMS.** The digestion of serpentine muds and the reference standard BHVO-2 followed a modified method of van den Boorn et al.[61]. About 30 mg of NaOHxH$_2$O (≥99.99% suprapur, Merck) were added to about 1 mg of pulverized serpentine mud and put on a hot plate by 100 °C for 3 days. Subsequently, the sample-NaOHxH$_2$O mix was re-dissolved in 1 ml Milli-Q (MQ) water and then centrifuged to separate undissolved solid material from the supernatant. The solid residue was separated from the supernatant and further processed with 100 μL 6 M HNO$_3$ on a hot plate for about 1 h. After the reaction, the sample was centrifuged and the solid residue separated from the supernatant. The residue was again mixed with about 30 mg of NaOHxH$_2$O, put on a hot plate at 100 °C for three days and subsequently 1 ml MQ water was added to re-dissolve the sample-NaOH mix. Yields are >97% and the procedure blank measured by ICP-OES is <1 μg.

Pore fluids and digested serpentine mud samples were purified following a method of Georg et al.[62]. The pH of the samples was adjusted to about 2 with concentrated HNO$_3$. One ml of the samples with a concentration of ~65 μM Si were loaded onto pre-cleaned cation-exchange resins (Biorad AG50 W-X8) and subsequently eluted with two ml MQ water.

Si isotopes of the pore fluids were measured on the NuPlasma HR MC-ICPMS at GEOMAR in medium resolution using the Cetac Aridus II desolvator. The intensity for 22 μM Si ranged between 3 to 4 V for $^{28}Si$ and the MQ blank was ≤3 mV (blank/signal ratio < 0.1%). Possible effects of organics and sulfate, which can significantly affect Si isotope measurements[63,64], were found to be negligible (see Ehlert et al.[39] for details). Si isotopes of the serpentinite mud samples were measured on the NeptunePlus HR MC-ICPMS at GEOMAR in medium-resolution mode and wet-plasma conditions, using a Teflon spray chamber. The instrumental mass bias was controlled by Mg doping of the purified samples[65,66]. The $^{28}Si$ intensity of a sample concentration of 35 μM yielded 2.5 V and the MQ blank was ≤20 mV (blank/signal ratio < 0.8%). All samples were measured using the standard-sample bracketing method to account for mass drifts of the instrument[67] and are reported in the $\delta^{30}Si$ notation, representing the deviation of the sample $^{30}Si/^{28}Si$ from that of the international Si standard NBS28 in permil (‰). NuPlasma HR MC-ICPMS long-term $\delta^{30}Si$ values of the reference materials Big Batch, IRMM018, BHVO-2, and Diatomite are −10.6 ± 0.2‰ (2 SD; n = 49), −1.5

± 0.2‰ (2 SD; n = 48), −0.3 ± 0.2‰ (2 SD; n = 13), and +1.3 ± 0.2‰ (2 SD; n = 44), respectively. In addition, an in-house pore fluid matrix standard has been measured which has an average $\delta^{30}Si$ value of +1.3 ± 0.2‰ (2 SD; n = 17) and the seawater inter-calibration standard Aloha (1000 m) resulted in +1.3 ± 0.2‰ (2 SD; n = 8), which is in very good agreement to Grasse et al.[68]. The reference materials measured at the NeptunePlus HR MC-ICPMS had $\delta^{30}Si$ values of −1.46 ± 0.09‰ (2 SD; n = 45) for IRMM018, +1.25 ± 0.10‰ (2 SD; n = 34) for Diatomite, and −0.33 ± 0.13‰ (2 SD; n = 12) for BHVO-2. The seawater inter-calibration standard Aloha (1000 m) had a $\delta^{30}Si$ values of +1.20 ± 0.07‰ (2 SD; n = 8). Re-measurements of some serpentine muds on the NuPlasma HR MC-ICPMS were identical within error to Si isotope results obtained at the NeptunePlus HR MC-ICPMS. All samples were measured 2–4 times on different days and the sample $\delta^{30}Si$ uncertainties ranged between 0.1 and 0.4‰ (2 SD, Supplementary Table 1).

**Boron sample preparation and isotope analyses by MC-ICPMS.** For boron isotope measurements, the pore fluids were purified using the microsublimation technique after a modified method by Gaillardet et al.[69]. The samples were diluted to 162 μM with HNO$_3$ and the pH adjusted to <2, so that all B was trigonal coordinated. Sample volumes of 30 μl were transferred on a lid of a conical Teflon vial, closed upside down, and put on a hot plate at 70 °C for 22 h. The Teflon vials were kept in a heating block to ensure uniform temperature distributions and condensation of the fluid in the cooler upper tip of the vial. Given that B is highly mobile during microsublimation, B was extracted from the remaining matrix and trapped in the vial, while the matrix remained at the lid. After opening the lid, the matrix and B were separated and the sample diluted with 1.47 ml of 0.5 M HNO$_3$ to measurement concentrations of 3.2 μM B. Boron recoveries were ≥97% and the total procedure blank <50 pg.

Boron isotopes were measured on the NeptunePlus HR MC-ICPMS in wet plasma using a Teflon spray chamber in low resolution mode. Details of the measurement set-up can be found in Jurikova et al.[70]. The samples were run using the standard-sample bracketing technique in order to account for machine-induced mass fractionation. NIST SRM 951 boric acid was used as bracketing standard. The $^{11}B$ intensities were ~1 V and the blank yielded intensities of ~20 mV, yielding sample/blank ratios ≤2%. Before and after each sample or standard, an on-peak zero of 0.5 M HNO$_3$ was measured containing the same fluid volume (1.5 ml) to ensure corrections of the B blank and B memories in the spray chamber during the measurement. Purified GEOTRACES seawater was used as matrix standard and yielded $\delta^{11}B$ values of +39.7 ± 0.4‰ (2 SD; n = 49). Pure boron reference materials[71] also processed by microsublimation showed $\delta^{11}B$ values of −20.5 ± 0.3‰ (2 SD; n = 6), +19.7 ± 0.2‰ (2 SD; n = 14), and +39.7 ± 0.3‰ (2 SD; n = 8) for ERM-AE120, ERM-AE121, and ERM-AE122, respectively. The long term (years 2016–2018) $\delta^{11}B$ values of pure boron reference materials[71] not processed by microsublimation were ERM-AE120 with −20.3 ± 0.3‰ (2 SD; n = 58), ERM-AE121 with +19.7 ± 0.2‰ (2 SD; n = 123), and ERM-AE122 with +39.7 ± 0.2‰ (2 SD; n = 104). The samples were measured 2-times per sequence on two to three days and had reproducibilities between 0.1 and 0.3‰ (2 SD; Supplementary Table 1).

**Strontium sample preparation and isotope analyses by TIMS.** The Strontium isotope ratios were measured by Thermal Ionization Mass Spectrometry (TIMS, Triton, ThermoFisher Scientific). The samples were chemically separated via cation exchange chromatography using the SrSpec resin (Eichrom). The isotope ratios were normalized to NIST SRM 987, which has a $^{87}Sr/^{86}Sr$ ratio of 0.710248[72] and the measurements yielded a precision of ±0.000015 (2 SD, n = 12).

**Mixing curve between seawater and a mantle-derived fluid.** The mixing curve between the two endmembers seawater and a mantle-derived fluid was calculated following:

$$\delta^{30}Si_{mix} = \frac{(\delta^{30}Si_{SW} \times [Si]_{SW} \times f) + (\delta^{30}Si_{mantle\,fluid} \times [Si]_{mantle\,fluid} \times (1-f))}{([Si]_{seawater} \times f) + ([Si]_{mantle\,fluid} \times (1-f))} \quad (1)$$

with $\delta^{30}Si_{sw}$ and $[Si]_{sw}$ as the respective seawater Si isotope composition (+1.05‰) and concentration (145.7 μM)[15], and $\delta^{30}Si_{mantle\,fluid}$ as the mantle-derived fluid Si isotope composition (+1.6‰; Supplementary Table 1, U1498B). The Si concentration of the mantle-derived fluid is below detection limit. Therefore, the Si concentration of a mantle-derived fluid from the nearby South Chamorro seamount was used (24 μM)[26]. Further Si concentrations of 5 μM and 70 μM were tested (gray-shaded area in Fig. 2) in order to account for the Si concentration range detected at nearby seamounts[26]. Mixing fractions are represented by f, varied over 100% seawater and 0% mantle-derived fluids and vice versa.

The same equation was used for $\delta^{11}B_{mix}$, with $\delta^{11}B_{sw}$ and $[B]_{sw}$ as the seawater B isotope composition (+39.6‰)[30] and concentration (432.6 μM)[29], respectively, and $\delta^{11}B_{mantle\,fluid}$ and $[B]_{mantle\,fluid}$ as the mantle-derived fluid B isotope composition (+16.1‰) and concentration (111 μM), respectively (Supplementary Table 1, U1498B).

**Numerical transport-reaction modeling.** A transport-reaction model was set up to simulate the processes occurring in the surface zone of serpentinite seamounts. The 1-D pore fluid model considers molecular diffusion, upward fluid flow, bottom

water intrusion in surface sediments, serpentine dissolution, and talc precipitation. The turnover of dissolved species in the pore fluid is simulated applying the following mass balance equation:

$$\Phi \times \frac{\partial C}{\partial t} = \frac{\partial}{\partial x}\left(\Phi \times \left(D_s \times \frac{\partial C}{\partial x} + v \times C\right)\right) + \Phi \times \alpha \times (C_{BW} - C) + \Phi \times R \quad (2)$$

with $\Phi$: porosity, $C$: concentration of dissolved species in the pore fluid ($\mu$mol cm$^{-3}$), $t$: time (yr), $x$: sediment depth (cm), $D_S$: molecular diffusion coefficient of dissolved species in sediment pore fluid (cm$^2$ yr$^{-1}$), $v$: upward fluid flow velocity of pore fluid (cm yr$^{-1}$), $\alpha$: mixing coefficient (yr$^{-1}$), $R$: turnover rates of dissolved species ($\mu$mol cm$^{-3}$ yr$^{-1}$). The model was set up for the following dissolved species: Si, $^{30}$Si, B, $^{11}$B, Sr, $^{87}$Sr, Cl. The isotopic compositions of the pore fluids ($\delta^{30}$Si, $\delta^{11}$B, $^{87}$Sr/$^{86}$Sr) are calculated from the corresponding mole fractions ($^{30}$Si/Si, $^{11}$B/B, $^{87}$Sr/Sr) applying previously published approaches[39,40] and the boundary conditions, equations, and parameter values are given in the Supplementary Methods.

## Data availability

All data generated or analyzed during this study are included in this published article (and its Supplementary Information files).

## Code availability

All algorithms used in this study are included in this published article (and its Supplementary Information files).

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

## Acknowledgements

We appreciate the support of the master and crew of the JOIDES Resolution during the IODP Expedition 366 Science Party. We thank Regina Surberg and Ana Kolevica for analytical support. Further thanks go to Jutta Heinze, Daphne Bartels, and Tyler Goepfert. IODP Germany and ECORD are thanked for financial support, as well as IODP grant code: NERC UK IODP Phase 2 Moratorium Award NE/P020909/1.

## Author contributions

C.D.M. was involved in research cruise IODP366 and carried out the sampling. S.G. and P.G. performed the Si isotope analyses. S.G. performed the B isotope analyses. V.L. conducted the Sr isotope measurements. K.W. designed the transport-reaction model. S.G., P.G., C.M., and K.W. interpreted the isotope data. S.G. wrote the manuscript with help from P.G., C.D.M., V.L., and K.W.

## Funding

## Competing interests

The authors declare no competing interests.
