## [Peer Review File · Nature Communications]

Reviewers' comments:

Reviewer #1 (Remarks to the Author):

Review of manuscript NCOMMS-19-35155 submitted to Nature Communications by Sonja Geilert and colleagues: "Serpentine alteration and the impact on the marine Si cycle"

Geilert et al. present the results of silicon and boron isotope analyses ($\delta^{30}\text{Si}$, $\delta^{11}\text{B}$) from porewaters and sediments overlying two seamounts where serpentinization is actively occurring. Serpentinization leaches Si from seafloor rocks, some fraction of which ultimately makes it to the ocean. This might thus be an underappreciated source of Si in the global budget, with potentially an exotic Si isotope composition. Sr isotopes tell them that fluids in most of their samples are of seawater origin. These span a range of $\delta^{30}\text{Si}$ and $\delta^{11}\text{B}$ values, which are attributed variously to clay dissolution (low $\delta^{30}\text{Si}/\delta^{11}\text{B}$) and clay formation (high values). They derive from a model an estimate of the Si isotope fractionation associated with serpentine alteration, and suggest that the high Si, high $\delta^{30}\text{Si}$ fluids produced may be meaningful for local and global Si cycling studies.

The manuscript is generally well written, largely appropriately referenced, and the figures clear. It is not a large dataset, but it is novel. My main criticism is that the interpretation of the results is rather descriptive and post-hoc and doesn't seem to produce a consistent framework that explains the both Si and B concentrations and isotope ratios. In particular, the interpretation of the (lack of) correlation between Si and B isotopes (L158 – L173) as reflecting different stages mafic rock/serpentine alteration is unconvincing.

The methods are adequately described, but analytically, it is concerning that the porefluids were directly loaded on to cation exchange columns: the Si elution fraction would then presumably include all other non- and negatively- charged solutes. Previous work has shown how e.g. sulfate or other anions in the solution can induce matrix effects during MC-ICP-MS. Could this be contributing to some of the extreme values measured?

If the study site is unique as a serpentinization site (L57), what does that mean for the representivity of the data and conclusions? In general, the manuscript lacks an assessment and discussion of how the fractionations derived are governed and applicable elsewhere (even assuming they are correct).

Temperature as a possible controlling factor of isotope fractionation is not considered. This seems particularly relevant given experimental and field evidence has demonstrated a relatively high degree of sensitivity to temperature changes in the ranges relevant here.

I believe the numerical modeling can be improved. I question if a Rayleigh model is the right approach, since this implies that a package of Si can only be progressively depleted, while it seems more likely that Si is both added and removed by different reactions simultaneously (indeed, this is even mentioned on L147). This invalidates the assumptions inherent in the Rayleigh model for isotope ratio evolution. In general, no motivation is given for choosing a Rayleigh model. Also, does it make sense to choose a 'closed system' model (S-L27)? In general, I don't find the modeling approach to calculate a fractionation (in the supplement) very convincing. How is the 'best fit scenario' derived and quantified? It seems like there should be many parameter assemblages that work well, and there is no need for each sample to be explained by the same model (or set of model parameters).

Given that the interpretation of data for samples from 1498B is different ('mantle sourced fluids' rather than low-T alteration processes) it might be worthwhile to also differentiate them in the figures.

If the secondary phases that are inferred to form are phyllosilicates, It's not clear to me how this process of 'serpentine alteration' is qualitatively different from authigenic clay formation observed or

interpreted elsewhere in different marine settings. I wonder if this body of literature, both Si specific and not, might help with some of the arguments made here.

Minor comments

L16: 'seawater alteration on the seafloor' is odd formulation, rephrase.

L45: Strange phrasing, "the dependence of Si isotope fractionation on..." or similar might be better.

L48/L51: Are these two statements contradictory? i.e. B is both incorporated into serpentine but is also incompatible?

L73/Table 1S: It seems the 'increasing Si concentrations with sediment depth' is a bit of an oversimplification.

L74: But some of the values are substantially below overlying seawater $\delta^{30}\text{Si}$ and concentration, which is unexpected and not explained. (i.e. the lowest $\delta^{30}\text{Si}$ value is also low Si concentration, which implies Si removal rather than addition as inferred). Conversely, some of the

L182: This value seems high, Gunnarsson and Arnorsson seem to report lower values at 8 C.

L207: These electronegativity arguments and the Meheut et al. paper cited refer to equilibrium isotope fractionation, which to my knowledge do not approach near the magnitude of -4.5‰, which requires a kinetic isotope effect (and no relation with Mg has yet been described.)

L240a: If river flux is ca. 6×10^{12} mol/yr, then this would be closer to 7% than 3%.

L240b: Why would this (empirical) estimate of ridge flank circulation, which is certainly crude, not already include serpentine alteration?

Table S1: Some samples have Si concentrations below detection: how is this possible within the framework advanced here?

Reviewer #2 (Remarks to the Author):

This manuscript tries to use Si isotopes of pore fluids from serpentine seamounts to study serpentinization in the mantle wedge and during seawater alteration on the seafloor, and explain anomalies in the marine Si budget. However, because the discussion is significantly flawed, this submission does not match the requirement for being published in Nature Communication.

First, I am not sure why it is necessary to report B isotopes in this manuscript. These data did not help explain any process in serpentinization or get the conclusion of this study. I am also curious why the authors try to use Si and B isotopes of pore fluids to decipher the formation of authigenic secondary phases in subducted slab?

Secondly, the manuscript did not discuss why the $\delta^{30}\text{Si}$ values of pore fluids increased with depth increasing. Higher temperature? Or different mineral phases? Is it possible that there is other mechanism to produce this heavy Si isotope signature in pore fluids? A deeper source, which can mix with pore fluids?

Also, a mass balance model is needed if trying to constrain the silicon fluxes of the global marine Si cycle.

The English of this manuscript should be improved.

Line 126-129:

Not really understand what the authors try to discuss here.

Line 145-147:

Why not? No evidence has been discussed here to prove that "pore fluid $\delta^{30}\text{Si}$ values cannot originate from serpentine dissolution alone".

Line 155-156:

The thin sections of drill samples will give direct evidence for the formation of authigenic secondary phases rather than both Si and B isotopes. It is not necessary to use these two isotope systems to decipher mineral phases in these drill core samples (pore fluids).

Line 165-166:

"B isotopes stay relatively stable at $\delta^{11}\text{B}$ values of +38‰ (Fig. 3a)". I cannot find the stable $\delta^{11}\text{B}$ values of +38‰ in Fig. 3a. There is only one spot of Yinazao sample has $\delta^{11}\text{B}$ value of +38.2‰. There are five samples with $\delta^{11}\text{B}$ values around +38‰ (+37.5 to +38.6‰), and their $\delta^{30}\text{Si}$ values varied from +0.4 to +2.7‰. However, the samples with high $\delta^{11}\text{B}$ values (15-R-1 and 3-F-2), their $\delta^{30}\text{Si}$ values are also in this range. It cannot be explained as "Instead, B is potentially trapped without species preferences in nanotubes within the crystal structure and $\delta^{11}\text{B}$ remains unfractionated". It is not appropriate to select some data to give an explanation, and leave the other data behind.

Response Letter

Kiel, 15.07.2020

Manuscript No.: NCOMMS-19-35155

Title: Serpentine alteration and the impact on the marine Si cycle

Dear Kyle Frischkorn,

Please find enclosed our revised manuscript '*Serpentine alteration and the impact on the marine Si cycle*'. We appreciate the comprehensive comments and suggestions for improvements made by the reviewers. The most significant change based on the reviewers' comments is the incorporation of a numerical transport-reaction model which can simulate and quantify the key mineral reactions involved. With the incorporation of the transport-reaction model, we have reduced the descriptive appearance of the first version of the manuscript and consolidated the statement of the manuscript, making it more impactful and relevant for publication in Nature Comms. The model outcome quantified the processes of mineral reactions associated with the extremely high isotope values and thus, the key message of the manuscript has not changed from its earlier version. We further emphasize the importance of using coupled Si and B isotope systems now in more detail (i.e. lines 60-63, 144-149, and 230-232) and have re-organized the figures to clarify the two main sources of the discussed fluids (seawater and mantle) and to illustrate the new model results. Further, we have deepened the discussion on why the results from the Mariana forearc are representative for the general oceanic crust (same mineralogy of authigenic minerals; section 1.3). We refrained from including a mass balance model, because the Si fluxes related to oceanic crust and serpentinite alteration by seawater are not well enough constrained and our new isotope data presented here is a significant contribution in itself, and therefore such a model is beyond the scope of this pioneering study. Please find below our replies (in blue) to the reviewer comments and changes in the manuscript in the attached marked-up version. Indicated line numbers refer to the revised manuscript.

We are pleased that the revised version of the manuscript has significantly improved and we hope that you agree and will consider this manuscript for publication.

With best regards on-behalf of all co-authors

Sonja Geilert

Reviewer #1 (Remarks to the Author):

Review of manuscript NCOMMS-19-35155 submitted to Nature Communications by Sonja Geilert and colleagues: "Serpentine alteration and the impact on the marine Si cycle"

Geilert et al. present the results of silicon and boron isotope analyses ($\delta^{30}\text{Si}$, $\delta^{11}\text{B}$) from porewaters and sediments overlying two seamounts where serpentinization is actively occurring. Serpentinization leaches Si from seafloor rocks, some fraction of which ultimately makes it to the ocean. This might thus be an underappreciated source of Si in the global budget, with potentially an exotic Si isotope composition. Sr isotopes tell them that fluids in most of their samples are of seawater origin. These span a range of $\delta^{30}\text{Si}$ and $\delta^{11}\text{B}$ values,

which are attributed variously to clay dissolution (low $\delta^{30}\text{Si}/\delta^{11}\text{B}$) and clay formation (high values). They derive from a model an estimate of the Si isotope fractionation associated with serpentine alteration, and suggest that the high Si, high $\delta^{30}\text{Si}$ fluids produced may be meaningful for local and global Si cycling studies.

The manuscript is generally well written, largely appropriately referenced, and the figures clear. It is not a large dataset, but it is novel. My main criticism is that the interpretation of the results is rather descriptive and post-hoc and doesn't seem to produce a consistent framework that explains the both Si and B concentrations and isotope ratios. In particular, the interpretation of the (lack of) correlation between Si and B isotopes (L158 – L173) as reflecting different stages mafic rock/serpentine alteration is unconvincing.

We agree that the interpretation of the isotope variations was not fully explained and not proven independently. Therefore, we included a one-dimensional transport-reaction model to constrain serpentine dissolution and authigenic mineral precipitation reactions and related isotope fractionation. We are aware that a 1-D transport-reaction model is not capable of resolving the complexity of the 3-D geometry of the seamounts, the time- and space dependencies of the mixing rates and variable fluid flow velocities. However, despite the simplified model used here, the geochemical and isotope data are well reproduced and processes within the seamount can be identified to be controlled by mineral dissolution and precipitation in contact with seawater and related changes in mineral solubility. Additionally, we added concentration data of Mg^{2+} and Cl^- as well as porosity to further constrain the model results (revised figures 3 and 4). The model set-up is briefly described in the main text and in more detail in the Method section and in the Supplement. The model results are discussed in lines 151-228 in the revised manuscript. Additionally, former figure 2 ($^{87}\text{Sr}/^{86}\text{Sr}$ versus $1/\text{Sr}$) was replaced by $^{87}\text{Sr}/^{86}\text{Sr}$ - and Sr concentration-depth plots to implement the model results (new figure 4).

The methods are adequately described, but analytically, it is concerning that the pore fluids were directly loaded on to cation exchange columns: the Si elution fraction would then presumably include all other non- and negatively- charged solutes. Previous work has shown how e.g. sulfate or other anions in the solution can induce matrix effects during MC-ICP-MS. Could this be contributing to some of the extreme values measured?

It is true that e.g. anions could affect Si isotope measurements. Therefore, we have tested beforehand in an earlier study (Ehlert et al., 2016) of Si isotope variations in marine pore fluids the effect of the presence of organics and SO_4 on the measurement outcomes, which were found to be the species, most likely influencing the Si isotope measurements (van den Boorn et al., 2009; Hughes et al., 2011; Oelze et al., 2016). The tests showed that the Si isotope results were identical within error if organics and SO_4 were removed or not. The test and the results are reported in Ehlert et al. (2016). Also the NuPlasma MC-ICPMS is more robust against anion effects compared to the Neptune MC-ICPMS, the latter overcomes this issue by Mg-doping (see below). We have now put a comment on this matter in the method section, in lines 600-601.

Furthermore, the spiking of Mg is additionally stabilizing the measurements on the Neptune MC-ICPMS and overcomes possible effects of anions during ionization (e.g. Oelze et al., 2016). This is mentioned in the Method section in lines 602-605.

If the study site is unique as a serpentinization site (L57), what does that mean for the representivity of the data and conclusions? In general, the manuscript lacks an assessment and discussion of how the fractionations derived are governed and applicable elsewhere (even assuming they are correct).

Despite the fact that the Mariana seamounts represent a unique serpentinization site, the controlling mineral reactions are comparable/ identical to mineral reactions occurring at the entire seafloor. The process of mafic and ultramafic mineral dissolution (basalt or serpentine alteration) and the subsequent re-precipitation of Si and other cations as authigenic phyllosilicates is reported from numerous seafloor sites (e.g. Iberian Abyssal Plain (Milliken et al., 1996), Juan de Fuca plate (Wheat and Mottl, 2000). Especially the authigenic mineral saponite is reported as the most common secondary mineral in altered oceanic crust (Coogan and Gillis, 2018) and as saponite also forms in the investigated Mariana seamounts (Morrow et al., 2019), we strongly argue for a transferability of the Mariana seamount results to general alteration reactions of the oceanic crust. We included the discussion of saponite formation in section 1.3 and clarified the transferability of the results from the Mariana seamounts to the oceanic crust in general in lines 258-275.

Temperature as a possible controlling factor of isotope fractionation is not considered. This seems particularly relevant given experimental and field evidence has demonstrated a relatively high degree of sensitivity to temperature changes in the ranges relevant here.

We agree that temperature changes can significantly affect isotope fractionation. However, taking the thermal gradients into account (Yinazao $\sim 20\text{-}30^\circ\text{C}/\text{km}$, Fantangisña summit $\sim 11.7^\circ\text{C}/\text{km}$; Fryer et al., 2018) the temperature change with sampling depth is only minor. At Yinazao, the deepest sample is located at 21 m below seafloor, which translates into a maximum temperature increase of 0.6°C (from 4°C seafloor to 4.6°C at 21 m depth using the highest measured thermal gradient of $30^\circ\text{C}/\text{km}$). The deepest sample at Fantangisña located 175 m below seafloor shows a temperature increase of 2°C (from 4 to 6°C). Temperature variations of $\leq 2^\circ\text{C}$ are to date not yet detectable in isotope fractionations and generally within measurement uncertainty (e.g. Geilert et al., 2014). Therefore, we exclude significant impact of temperature variations with sampling depth on the Si and B isotope variability in studied pore fluids. We added a short sentence noting this in lines 78-79.

I believe the numerical modeling can be improved. I question if a Rayleigh model is the right approach, since this implies that a package of Si can only be progressively depleted, while it seems more likely that Si is both added and removed by different reactions simultaneously (indeed, this is even mentioned on L147). This invalidates the assumptions inherent in the Rayleigh model for isotope ratio evolution. In general, no motivation is given for choosing a Rayleigh model. Also, does it make sense to choose a 'closed system' model (S-L27)? In general, I don't find the modeling approach to calculate a fractionation (in the supplement) very convincing. How is the 'best fit scenario' derived and quantified? It seems like there should be many parameter assemblages that work well, and there is no need for each sample to be explained by the same model (or set of model parameters).

We have substantially revised the model section of the manuscript in lines 151-228 of the revised manuscript (see also comment to remark #1). We agree that a Rayleigh model does not account for the complexity and duality of processes occurring and was mainly included

to illustrate a simplified process of authigenic mineral precipitation. The Rayleigh model and the closed-system numerical model were removed from the revised manuscript and instead a numerical reactive transport-reaction model (steady state conditions) was included. Sensitivity tests concerning the magnitude of Si isotope fractionation are included in the model to constrain the associated $\delta^{30}\text{Si}$ values. The $\delta^{30}\text{Si}$ values can only be reproduced in the model with a Si isotope fractionation of -3‰ (see Fig. 6c; lines 184-187).

Given that the interpretation of data for samples from 1498B is different ('mantle sourced fluids' rather than low-T alteration processes) it might be worthwhile to also differentiate them in the figures.

The symbols of the mantle-sourced fluid were changed to better differentiate them from the seawater-altered fluids.

If the secondary phases that are inferred to form are phyllosilicates, It's not clear to me how this process of 'serpentine alteration' is qualitatively different from authigenic clay formation observed or interpreted elsewhere in different marine settings. I wonder if this body of literature, both Si specific and not, might help with some of the arguments made here.

The alteration of mafic and ultramafic rocks and the formation of authigenic phyllosilicates is occurring in the Mariana seamounts and the results are transferrable to the entire oceanic crust as we have argued above (comment to reviewer remark #3 'Especially the authigenic mineral saponite is reported as the most common secondary mineral in altered oceanic crust (Coogan and Gillis, 2018) and as saponite also forms in the investigated Mariana seamounts (Morrow et al., 2019), we strongly argue for a transferability of the Mariana seamount results to general alteration reactions of the oceanic crust.'). We agree that the transferability might not have been stated clearly in the original manuscript and we have revised the section in lines 258-275. Additionally, we added a short paragraph in which we compare the mineral reactions occurring in the Mariana seamounts to authigenic mineral formation in continental margin settings (lines 194-209).

Minor comments

L16: 'seawater alteration on the seafloor' is odd formulation, rephrase.

The sentence was rephrased.

L45: Strange phrasing, "the dependence of Si isotope fractionation on..." or similar might be better.

The sentence was rephrased accordingly.

L48/L51: Are these two statements contradictory? i.e. B is both incorporated into serpentine but is also incompatible?

This is not a contradiction – although B prefers the liquid phase ($D(s/f)=0.25$), a lot of B can also be stored in the crystal lattice of the serpentine compared to other fluid mobile

elements (Li, Cs, Sr), meaning that B-rich fluids precipitating serpentine provide B-rich serpentine. The large amount of B capable of being incorporated into serpentine is relative to other minerals, but B still remains incompatible (just less incompatible with serpentine than with other minerals). Its K_d between water and mineral is less than 1 (~ 0.25 , Tenhorey and Hermann, 2004), but higher than many other incompatible elements. When fluids have very high B concentrations there is a high capability for serpentine to incorporate B into its structure. We included a short comment in lines 57-59.

L73/Table 1S: It seems the 'increasing Si concentrations with sediment depth' is a bit of an oversimplification.

We agree that the description was an over simplification of the profiles and modified the text accordingly (lines 86-87).

L74: But some of the values are substantially below overlying seawater $\delta^{30}\text{Si}$ and concentration, which is unexpected and not explained. (i.e. the lowest $\delta^{30}\text{Si}$ value is also low Si concentration, which implies Si removal rather than addition as inferred). Conversely, some of the

The statement that most of the $\delta^{30}\text{Si}$ values are above seawater is valid (10 out of 13 samples have higher $\delta^{30}\text{Si}$ values compared to seawater). The $\delta^{30}\text{Si}$ values lower than seawater $\delta^{30}\text{Si}$ are explained now at the end of section 1.2.2 (lines 232-244) and are likely caused by different mineral solubilities related to the ambient fluid. In order to cause these low pore fluid $\delta^{30}\text{Si}$ values, mineral phases with low isotopic values need to dissolve. We speculate that the most likely mineral phases are earlier formed authigenic clays. The authigenic clays likely formed during higher ascent rates of the deep mantle-derived fluids in the past causing oversaturation of these phases and precipitation. At present, the ascent rates likely decreased shifting the dominant fluid phase to seawater, which causes re-dissolution of these phases. Significant variabilities between ascent rates were observed between the different seamounts in the Mariana forearc region (between 2 mm yr^{-1} to 4 cm yr^{-1} ; Mottl et al., 2004) and thus changes over time are not unlikely. Further investigations are necessary to constrain thoroughly the fluid ascent rates and possible changes over time. However, the complex mixing of fluids with different Si concentrations (close to zero in the deep mantle and $\sim 140 \mu\text{M}$ in seawater) and simultaneously occurring Si mineral dissolution and re-precipitation likely explains the Si concentration ranges and $\delta^{30}\text{Si}$ variability in the studied samples.

L182: This value seems high, Gunnarsson and Arnórsson seem to report lower values at 8 C.

Gunnarsson and Arnórsson (2000) report an amorphous silica solubility of $\sim 80 \text{ ppm}$ (2.8 mM , reported as 3 mM in the main text) at 8°C . However, this part was removed from the revised manuscript and replaced with the results and interpretation from the reactive transport model (lines 151-228).

L207: These electronegativity arguments and the Meheut et al. paper cited refer to equilibrium isotope fractionation, which to my knowledge do not approach near the

magnitude of -4.5‰, which requires a kinetic isotope effect (and no relation with Mg has yet been described.)

We agree that the Méheut and Schauble (2014) refers to equilibrium fractionation and the reference was removed from the manuscript.

L240a: If river flux is ca. 6×10^{12} mol/yr, then this would be closer to 7% than 3%.

The percentage of Si derived from low-temperature weathering relative to the riverine flux is taken from Wheat et al. (2017), who based their calculation on riverine fluxes by Elderfield and Schultz (1996) and additionally took the global heat loss and thermal anomalies into account. However, we have revised this section given the large range of Si fluxes related to low-temperature oceanic crust weathering reported in literature (e.g. Wheat and McManus, 2005). In order to fully assess and model the Si flux from oceanic crust alteration, better constraints on alteration rates and related Si fluxes from various marine settings is required.

L240b: Why would this (empirical) estimate of ridge flank circulation, which is certainly crude, not already include serpentine alteration?

The reported ridge flank Si fluxes originate from a fluid discharged from altered basaltic crust (e.g. Juan de Fuca Plate (Wheat and Mottl, 2000)). To the best of our knowledge, no Si fluxes related to serpentine alteration are reported. It is likely, that the general fluid flux has similar magnitudes as derived from seafloor basalt alteration studies (in the range of 0.02 to 0.8 Tmol Si yr⁻¹; Wheat and McManus, 2005), but the associated Si concentration remains unknown. We rearranged this part of the discussion in lines 258-281.

Table S1: Some samples have Si concentrations below detection: how is this possible within the framework advanced here?

The reported Si concentrations were removed from Table S1.

Reviewer #2 (Remarks to the Author):

This manuscript tries to use Si isotopes of pore fluids from serpentine seamounts to study serpentinization in the mantle wedge and during seawater alteration on the seafloor, and explain anomalies in the marine Si budget. However, because the discussion is significantly flawed, this submission does not match the requirement for being published in Nature Communication.

First, I am not sure why it is necessary to report B isotopes in this manuscript. These data did not help explain any process in serpentinization or get the conclusion of this study. I am also curious why the authors try to use Si and B isotopes of pore fluids to decipher the formation of authigenic secondary phases in subducted slab?

The combination of Si and B isotopes helped to reveal the processes during seawater alteration of serpentine and secondary mineral precipitation. A large number of publications have shown that Si isotopes are a good tracer for alteration reactions, however, Si isotopes

have never been measured in relation to serpentine, in contrast to B isotopes. B isotopes are well studied to trace primary serpentinization reactions as well as secondary alteration reactions. Thus, the combination of a new tracer (Si isotopes) with a well-established tracer (B isotopes) provides the possibility to establish Si isotopes as a new proxy to trace serpentine alteration reactions.

In this manuscript, we show for the first time the very good correlation of both isotope systems during alteration reactions in that the light isotope is incorporated in the secondary mineral for both isotope systems. We have revised former section 1.2.3 (revised section 1.2.2) to clarify this coupling by incorporating a transport-reaction model using both isotope systems together with further geochemical data to identify and disentangle mineral reactions (lines 151-228 in the revised manuscript). We can show that processes within the seamount are controlled by serpentine dissolution and precipitation of authigenic minerals in contact with seawater and related changes in mineral solubility. The combination of Si and B isotopes has thus the potential to identify different alteration reactions during seafloor weathering.

Authigenic phases in the subducted slab, as mentioned by the reviewer, are not discussed in this manuscript. Si and B isotopes from deep mantle-derived fluids, discussed in section 1.2.1, relate to primary serpentinization process.

Secondly, the manuscript did not discuss why the $\delta^{30}\text{Si}$ values of pore fluids increased with depth increasing. Higher temperature? Or different mineral phases? Is it possible that there is other mechanism to produce this heavy Si isotope signature in pore fluids? A deeper source, which can mix with pore fluids?

We agree that the variations in pore fluid $\delta^{30}\text{Si}$ and $\delta^{11}\text{B}$ values were not sufficiently discussed in the manuscript. We have revised former section 1.2.3 (new section 1.2.2) by including a transport-reaction model which takes the ascent of a deep mantle-derived fluid into account that mixes with seawater in the upper 30m of the seamounts (lines 151-228 in the revised manuscript).

The geochemical and isotopic data are well reproduced by the model and alteration processes can be related to changing mineral solubilities and associated dissolution and precipitation reactions. Concentration data of Mg^{2+} , Sr, and Cl^- as well as $^{87}\text{Sr}/^{86}\text{Sr}$ and porosity were used to further constrain the model results (revised figures 3 and 4). See also answer to comment #1 of reviewer #1.

Temperature should only have a minor effect on the isotope data of the seawater-derived fluids in that thermal gradients cause only a temperature difference of $\leq 2^\circ\text{C}$ between the shallowest and deepest sample (Fryer et al., 2018). We included a comment in the main text in lines 78-79.

Also, a mass balance model is needed if trying to constrain the silicon fluxes of the global marine Si cycle.

We refrained from calculating a mass balance model for the global marine cycle, because we provide the first Si, B (isotope) data related to serpentine alteration. We strongly argue that the mineralogical changes involved in serpentine alteration are transferable to basalt alteration (see revised section 1.3, lines 258-275), however, this needs to be proven in further studies. In order to set-up a realistic seafloor alteration model further isotopic constraints are required to test this hypothesis. Additionally, reported Si fluxes related to

seafloor basalt alteration show a large range (e.g. 0.02 to 0.8 Tmol Si yr⁻¹; Wheat & McManus, 2005) and are thus not well enough constrained to justify a mass balance model. Nevertheless, we provide the first Si and B isotope data, which indicates that seafloor alteration reactions have the potential to influence oceanic Si budgets and hopefully inspire further studies to study this process.

The English of this manuscript should be improved.

The English of the manuscript was completely revised by an English native speaker.

Line 126-129:

Not really understand what the authors try to discuss here.

This section discusses the low isotopic values detected in the top 5 m of the seamounts. The low B and Si isotope values are related to the dissolution of a mineral phase with low $\delta^{30}\text{Si}$ values, most likely earlier formed authigenic clays. This section was revised taking into account the model results and is now part of section 1.2.2, lines 232-244. We speculate that the authigenic clays likely formed during interaction with the deep mantle-derived fluids in the past caused by higher fluid ascent rates. With respect to the deep fluids, these phases are oversaturated in the fluids and thus precipitate. At present, lower ascent rates of the deep fluid likely cause seawater entrainment and the earlier formed authigenic phases dissolve again due to their undersaturation in seawater (see also the revised Fig. 5 and 6). Ascent rates can vary significantly between seamounts in the Mariana forearc region (between 2 mm yr⁻¹ to 4 cm yr⁻¹; Mottl et al., 2004) and thus changes over time are not unlikely as well. Further investigations are necessary to constrain thoroughly the fluid ascent rates and possible changes over time.

Line 145-147:

Why not? No evidence has been discussed here to prove that “pore fluid $\delta^{30}\text{Si}$ values cannot originate from serpentine dissolution alone”.

We conclude that the pore fluid $\delta^{30}\text{Si}$ values cannot be generated by serpentine dissolution alone, because the pore fluids should then have the Si isotope signature from the serpentine, that is to say -0.3‰. Therefore, we conclude that another process must occur, shifting pore fluid $\delta^{30}\text{Si}$ to the observed high values. The most obvious process is Si re-precipitation and with that the precipitation of the light ^{28}Si , enriching the fluid in ^{30}Si . In order to clarify this process, we added a short paragraph in lines 173-176.

Line 155-156:

The thin sections of drill samples will give direct evidence for the formation of authigenic secondary phases rather than both Si and B isotopes. It is not necessary to use these two isotope systems to decipher mineral phases in these drill core samples (pore fluids).

We agree that investigations of thin sections will give direct evidence on the formation of authigenic phases and that the sentence might have been misleading. The main intention of studying both isotope systems was the possibility to trace mineral reactions and establish Si as a new tracer for alteration reactions at the seafloor. Often, effusing fluids from basement rocks are easier to access compared to basement rocks themselves. We show that

information on mineral reactions can be drawn from Si and B isotope variations in these fluids and that these isotope systems can be used to trace reactions at depth. We revised section 1.3 accordingly.

Line 165-166:

“B isotopes stay relatively stable at $\delta^{11}\text{B}$ values of +38‰ (Fig. 3a)”. I cannot find the stable $\delta^{11}\text{B}$ values of +38‰ in Fig. 3a. There is only one spot of Yinazao sample has $\delta^{11}\text{B}$ value of +38.2‰. There are five samples with $\delta^{11}\text{B}$ values around +38‰ (+37.5 to +38.6‰), and their $\delta^{30}\text{Si}$ values varied from +0.4 to +2.7‰. However, the samples with high $\delta^{11}\text{B}$ values (15-R-1 and 3-F-2), their $\delta^{30}\text{Si}$ values are also in this range. It cannot be explained as “Instead, B is potentially trapped without species preferences in nanotubes within the crystal structure and $\delta^{11}\text{B}$ remains unfractionated”. It is not appropriate to select some data to give an explanation, and leave the other data behind.

Former section 1.2.3 (revised section 1.2.2) was completely revised to clarify the mineral reactions during serpentine alteration. We removed the Rayleigh and closed-system model and instead included a transport-reaction model, which is briefly explained in the main text and a full description can be found in the Method section and the Supplement. We can now show that the alteration reactions are controlled by mineral solubilities in that serpentine dissolves in the upper 30m of the seamount and secondary minerals like talc reach oversaturation and precipitate, thereby incorporating the light isotopes of both investigated systems and shifting fluid $\delta^{30}\text{Si}$ and $\delta^{11}\text{B}$ to the observed high values. We added new figures to illustrate the model results together with the measured geochemical data (revised Fig. 2 - 6) and discuss the model results in lines 151-228.

Review #3 of Serpentine alteration and the impact on the marine Si cycle

Authors Sonja Geilert, Patricia Grasse, Klaus Wallmann, Volker Liebetrau, Catriona D. Menzies

General Comments

I think this is a good paper worth publishing in Nature communications after some revisions and moderate changes. The Si isotope data is a novel approach to understand the role of serpentinites in the Si global cycle and along B provide good insight in the formation and nature of serpentine mud volcanoes and its potential to alter Si signatures in the ocean.

My main criticism to the paper is that it tries to expand the findings from the Mariana's trench serpentinite mud volcanoes to most of oceanic crust alteration without providing evidence for this link. While their findings are robust for serpentinite alteration, they do not have supporting evidence for basaltic crust. Serpentinization reactions in peridotites are very different from basalt alteration reactions. I think it is fine for the authors to hypothesize that Si isotopes during basalt alteration might behave similarly and further study is required as their data do not directly apply to basalt alteration. I think this needs to be modified.

We agree that we can only speculate about the Si isotope signature of fluids related to basalt alteration. Here, we want to show that serpentine alteration (not formation!) is comparable to basalt alteration given that similar reaction products are formed. Reaction products for Si phases are mainly phyllosilicates like saponite, a Mg-rich smectite. We hypothesize that

similar mineralogical reactions during alteration will result in similar isotopic fractionations and that is why we think, that the results from serpentine alteration are transferable to basalt alteration. We rearranged the discussion concerning this part to clarify the similarity of authigenic minerals (section 1.3, lines 258-271) and defined the meaning of serpentine alteration in line 34-38, to avoid confusion with preceding serpentine formation.

Specific comments

Line 26 It is unclear who or why serpentinites are expected to play a fundamental role in Si exchange, is there a reference for this or is this the authors interpretation?

We included the reference Frost & Beard (2007) who discuss in detail the role of Si and silica activity in a serpentinizing environment (lines 27-29).

Line 40-44 Authors do not mention the variability in $\delta^{30}\text{Si}$ in seawater. Review by Poitrasson (2017) show seawater variability in $\delta^{30}\text{Si}$ from +0.5 to +4.0‰. Part of this variability is associated geographic location and some with depth see Holtzer and Brzezinski (2015). This moves to line 75 where authors compare pore fluids with NW Pacific seawater. The variability needs to be mentioned in the text and point that at depth is relatively constant in the Pacific.

We added a sentence of the $\delta^{30}\text{Si}$ variability in in the Pacific in lines 46-49 and emphasize the homogeneity of Pacific Ocean deep waters. We refer to Reynolds et al. (2026), Grasse et al. (2013) and the review by Sutton et al. (2018), that provides the latest data compilation. We have focused the discussion in the main text on Pacific $\delta^{30}\text{Si}$ values, given that the study area is located within the NW Pacific. We show that surface waters from the photic zone can have high $\delta^{30}\text{Si}$ values, up to 4.4‰, but that the deep waters are relatively homogeneous with an average $\delta^{30}\text{Si}$ of $+1.2 \pm 0.2\%$ (Grasse et al., 2013).

Line 50-51 Authors should add $\delta^{11}\text{B}$ data from Martin et al. 2016 to their range and incorporate into their discussion. Martin et al. use $\delta^{11}\text{B}$ to discriminate serpentinization environments and expand the range measured to serpentinites to lower values ($\delta^{11}\text{B} \sim -10\%$).

The data and reference of Martin et al. (2016) was added to the reported range of serpentine $\delta^{11}\text{B}$ values in line 57.

Line 81-83 If it is within analytical uncertainty, I recommend this sentence to be removed from the manuscript.

The reported range in $\delta^{30}\text{Si}$ for the serpentinite muds is outside the analytical uncertainty, which is reported in the method section (long-term uncertainty $\leq 0.13\%$; sample uncertainty between 0.1 and 0.4‰). Therefore, we will leave the reported range in $\delta^{30}\text{Si}$ values in the manuscript.

Line 103 Add B concentrations as done for $\delta^{11}\text{B}$.

The B concentrations were added in line 131.

Line 103-109 I suggest these lines are reorganized. The fact that fluids and clasts overlap is solid and points to no fractionation. I suggest that this is linked and then the authors explain their interpretation for no fractionation (ie B coordination).

The lines concerning deep-mantle serpentinization and the impact on $\delta^{11}\text{B}$ were reorganized accordingly (lines 126-135).

Line 110 Value of $\delta^{30}\text{Si}$ in pore fluids is close to seawater which is opposite to what is seen with Sr. Serpentinization fluids would have Si concentration an order of magnitude lower than seawater at 300C (see Klein et al. 2009) so I am worried that this is reflecting minute amounts of SW. Though lost city seawater has lower Si so this might not be relevant

We agree that the $\delta^{30}\text{Si}$ values are likely affected by shallow overprinting and do not represent deep serpentinizing fluids as stated in the original manuscript. The silica concentration is about a magnitude lower compared to seawater (<40 μM (below detection limit) versus 145 μM Si, Table S1), however, the $\delta^{30}\text{Si}$ value is higher than seawater (1.6‰ versus 1.05‰; Reynolds et al., 2006). This indicates that it is a deep, Si-depleted fluid as expected from Sr isotopic values, but Si precipitation processes shifted the $\delta^{30}\text{Si}$ values to higher values. We can only speculate that the precipitation process occurred close to the surface when temperature and pressure drop (see lines 135 to 142 in the revised manuscript). So far, due to the extremely low Si concentrations in serpentinizing fluids, no comparable $\delta^{30}\text{Si}$ values exist providing evidence for this hypothesis.

Line 119 If closed system dissociation would not matter much would it?

Also in a closed system, when Si dissociates at $\text{pH} > 8.5$, the precipitating phase would be enriched in ^{30}Si , shifting fluid $\delta^{30}\text{Si}$ to low values. Thus, if a pH effect on Si fractionation would occur than fluid $\delta^{30}\text{Si}$ values should be below its initial $\delta^{30}\text{Si}$ values, which is that of olivine ($\sim -0.3\text{‰}$). Given that $\delta^{30}\text{Si}$ values are higher than that of olivine, we exclude a pH-induced fractionation and assume a kinetically-controlled fractionation in dependence of decreasing T and P after ascent and deposition of the serpentinite muds (see lines 135 to 142 in the revised manuscript).

Line 130 B increase in the pore fluid?

Adsorbed B is released into the pore fluid during interaction with seawater, thus increasing B concentrations. This section was revised and is now part of section 1.2.2, lines 232-244.

Line 130-133 I recommend adding a reference for the borate fractionation. Apart from being the uppermost part of the seamounts are pelagic clays and silts related to the serpentinite muds?

A reference for boric acid-borate fractionation was added (lines 211-216).

The pelagic clays and silts form the uppermost layer of the mud volcanoes and are not directly related to the serpentinite muds and the process of serpentinization in general. For Yinazao seamount, pelagic muds are only reported in the upper about 5 m of the seamount (Fryer et al., 2018). At Fantangisña seamount, pelagic clays dominate at the top and at the base of the seamount with only few layers intersecting the serpentinite muds (about $\leq 10\%$;

Fryer et al., 2018). Therefore, pelagic clays play only a subordinate role in the investigated core sections and will not control the Si isotope signatures at depths > 5m.

Line 143 Need clarification for serpentinites. Are the authors referring to clasts brought up by the fluids or only serpentinite muds.

We refer to all serpentine minerals either in clasts as pervasive serpentine or as serpentine veins or in the serpentinized mud itself. When seawater reacts with serpentinite, serpentine minerals dissolves through hydrolysis following Milliken et al. (1996):

We included a short explanation as well as the reaction in lines 166-172.

Figure 3b add temperature of solubility calculations to figure or to caption.

This figure was deleted from the manuscript and replaced by the results of the transport-reaction model (revised figures 3-6). A brief description of the model can be found in the main text (lines 151-155) and a detailed description in the Method section and the Supplement. See also comment to reviewer #1 comment #1.

Line 242 I agree with the associated signature from serpentinites but not from basalts. I think it is fine for the authors to hypothesize that basalt alteration might be similar and further study is required but their data do not directly apply to basalt alteration and this needs to be modified.

We have revised the discussion in section 1.3 to clarify why serpentine alteration is comparable to basalt alteration (similar mineralogy of authigenic products). See also comment to 'Main criticism'.

References on this review

- Poitrasson, F., 2017. Silicon Isotope Geochemistry. *Rev. Mineral. Geochemistry* 82, 289–344. <https://doi.org/10.2138/rmg.2017.82.8>
- Holzer, M., Brzezinski, M.A., 2015. Controls on the silicon isotope distribution in the ocean: New diagnostics from a data-constrained model. *Global Biogeochem. Cycles* 29, 267–287. <https://doi.org/10.1002/2014GB004967>
- Martin, C., Flores, K.E., Harlow, G.E., 2016. Boron isotopic discrimination for subduction related serpentinites. *Geology* 44, 899–902. <https://doi.org/10.1130/G38102.1>

Juan Carlos de Obeso

References

- van den Boorn S. H. J. M., Vroon P. Z. and van Bergen M. J. (2009) Sulfur-induced offsets in MC-ICP-MS silicon-isotope measurements. *J. Anal. At. Spectrom.* **24**, 1111.
- Coogan L. A. and Gillis K. M. (2018) Low-Temperature Alteration of the Seafloor: Impacts on Ocean Chemistry. *Annu. Rev. Earth Planet. Sci.* **46**, 21–45.
- Ehlert C., Doering K., Wallmann K., Scholz F., Sommer S., Grasse P., Geilert S. and Frank M. (2016) Stable silicon isotope signatures of marine pore waters – Biogenic opal dissolution versus authigenic clay mineral formation. *Geochim. Cosmochim. Acta* **191**, 102–117. Available at: <http://dx.doi.org/10.1016/j.gca.2016.07.022>.
- Elderfield H. and Schultz a. (1996) Mid-Ocean Ridge Hydrothermal Fluxes and the Chemical Composition of the Ocean. *Annu. Rev. Earth Planet. Sci.* **24**, 191–224.
- Fryer P., Wheat C. G., Williams T. and Expedition 366 Scientists T. (2018) Mariana Convergent Margin and South Chamorro Seamount. *Proc. Int. Ocean Discov. Program, 366 Coll. Station. TX (International Ocean Discov. Program)*. Available at: <https://doi.org/10.14379/iodp.proc.366.2018>.
- Geilert S., Vroon P. Z., Roerdink D. L., Cappellen P. Van and van Bergen M. J. (2014) Silicon isotope fractionation during abiotic silica precipitation at low temperatures: inferences from flow-through experiments. *Geochim. Cosmochim. Acta* **142**, 95–114. Available at: <http://linkinghub.elsevier.com/retrieve/pii/S0016703714004542>.
- Gunnarsson I. and Arnórsson S. (2000) Amorphous silica solubility and the thermodynamic properties of H_4SiO_4^* in the range of 0° to 350°C at P(sat). *Geochim. Cosmochim. Acta* **64**, 2295–2307.
- Hughes H. J., Delvigne C., Korntheuer M., de Jong J., André L. and Cardinal D. (2011) Controlling the mass bias introduced by anionic and organic matrices in silicon isotopic measurements by MC-ICP-MS. *J. Anal. At. Spectrom.* **26**, 1892. Available at: <http://xlink.rsc.org/?DOI=c1ja10110b>.
- Méheut M. and Schauble E. A. (2014) Silicon isotope fractionation in silicate minerals : Insights from first-principles models of phyllosilicates, albite and pyrope. *Geochim. Cosmochim. Acta* **134**, 137–154. Available at: <http://dx.doi.org/10.1016/j.gca.2014.02.014>.
- Milliken K. L., Lynch F. L. and Seifert K. E. (1996) MARINE WEATHERING OF SERPENTINITES AND SERPENTINITE BRECCIAS , SITES 897 AND 899, IBERIA ABYSSAL PLAIN. In *Proceedings of the Ocean Drilling Program* (eds. R. B. Whitmarsh, D. S. Sawyer, A. Klaus, and D. G. Masson). pp. 529–540.
- Morrow C. A., Moore D. E., Lockner D. A. and Bekins B. A. (2019) Data report: permeability, porosity, and frictional strength of core samples from IODP Expedition 366 in the Mariana forearc. In *Fryer, P., Wheat, C.G., Williams, T., and the Expedition 366 Scientists, Mariana Convergent Margin and South Chamorro Seamount. Proceedings of the International Ocean Discovery Program, 366: College Station, TX (International Ocean Discovery Program)* Available at: <https://doi.org/10.14379/iodp.proc.366.202.2019>.
- Mottl M. J., Wheat C. G., Fryer P., Gharib J. and Martin J. B. (2004) Chemistry of springs across the Mariana forearc shows progressive devolatilization of the subducting plate. *Geochim. Cosmochim. Acta* **68**, 4915–4933.
- Oelze M., Schuessler J. A. and von Blanckenburg F. (2016) Mass bias stabilization by Mg doping for Si stable isotope analysis by MC-ICP-MS. *J. Anal. At. Spectrom.* **31**, 2094–2100. Available at: <http://xlink.rsc.org/?DOI=C6JA00218H>.
- Tenthorey E. and Hermann J. (2004) Composition of fluids during serpentinite breakdown in subduction zones: Evidence for limited boron mobility. *Geology* **32**, 865–868.
- Wheat C. G., Fisher A. T., Mcmanus J., Hulme S. M. and Orcutt B. N. (2017) Cool seafloor hydrothermal springs reveal global geochemical fluxes. *Earth Planet. Sci. Lett.* **476**, 179–188.
- Wheat C. G. and McManus J. (2005) The potential role of ridge-flank hydrothermal systems on oceanic germanium and silicon balances. *Geochim. Cosmochim. Acta* **69**, 2021–2029.
- Wheat C. G. and Mottl M. J. (2000) Composition of pore and spring waters from Baby Bare : Global implications of geochemical fluxes from a ridge flank hydrothermal system. *Geochim. Cosmochim. Acta* **64**, 629–642.

Serpentine alteration and the impact on the marine Si cycle

Sonja Geilert^{1*}, Patricia Grasse^{1,2}, Klaus Wallmann¹, Volker Liebetrau¹, Catriona D. Menzies^{3,2}

¹GEOMAR Helmholtz Centre for Ocean Research Kiel, Wischhofstr. 1-3, 24148 Kiel, Germany

²German Centre for Integrative Biodiversity Research (iDiv) Halle-Jena-Leipzig, Deutscher Platz 5e, 04103 Leipzig, Germany

^{3,2}Department of Earth Sciences, Durham University, Science Laboratories, South Road, Durham, UK

Keywords: Mariana seamounts, serpentine alteration, ~~large-Si isotopes, and~~ B isotopes ~~fractionation~~, authigenic clay formation, marine Si cycle

Abstract

Serpentine ~~ocean-floor~~ alteration is recognized as an important process for element cycling, however, related silicon fluxes are unknown. Pore fluids from serpentine seamounts sampled in the Mariana forearc region during IODP Expedition 366 were investigated for their Si, B, and Sr isotope signatures ($\delta^{30}\text{Si}$, $\delta^{11}\text{B}$, and $^{87}\text{Sr}/^{86}\text{Sr}$, respectively) to study serpentinization in the mantle wedge and ~~during shallow serpentine alteration to authigenic clays by~~ seawater ~~alteration on the seafloor~~. While serpentinization in the mantle wedge caused no significant Si isotope fractionation, implying ~~almost~~ closed system conditions, serpentine alteration by seawater led to the formation of ~~secondary authigenic~~ phyllosilicates, causing the highest natural fluid $\delta^{30}\text{Si}$ values measured to date (up to ~~+5.2±0.2‰~~). Seafloor alteration of serpentinites is a source of Si to the ocean with extremely high fluid $\delta^{30}\text{Si}$ values, which can explain anomalies in the marine Si budget like in the Cascadia Basin and which has to be considered in future investigations of the global marine Si cycle.

Serpentinites are expected to play a fundamental role in the exchange of Si between the Earth's mantle and the global ocean ~~since dissolved Si is taken up during serpentine formation~~¹. Serpentine is a hydrated Mg-silicate ($\text{Mg}_6\text{Si}_4\text{O}_{10}(\text{OH})_8$) which forms during the reaction of seawater with mantle rocks and occurs in a variety of marine settings including slow-spreading mid-ocean ridges, rifted continental margins, and fore-arc regions of subduction zones under a wide range of temperature and pressure regimes²⁻⁶. At slow and

34 ultra-slow spreading ridges, serpentinites can make up to 20% of the seafloor, extending 3-4
35 km into the footwall as seawater circulates through fractures and faults^{6,7}. Despite the
36 increasing awareness of the importance of serpentinization influencing global element
cycles, serpentine alteration (here defined as weathering reaction after preceding
serpentinization) during water-rock interactions at low temperatures (< 20°C) is not well
understood. During alteration reactions, Si is removed from mafic or ultramafic rocks and
partially re-precipitates as authigenic clay minerals, nevertheless resulting in a net gain of Si
to the ocean⁸. However, this Si flux is associated with an unknown Si isotope composition
($\delta^{30}\text{Si}$) adding a large uncertainty to models simulating the global Si cycles. Dissolution and
precipitation processes have been found to induce significant Si isotope fractionation during
biotic and abiotic processes (see recent reviews by Frings et al.⁹ and Sutton et al.¹⁰),
spanning a natural range in solid phase $\delta^{30}\text{Si}$ values between -5.7‰ (silcretes)¹¹ and +6.1‰
(phytoliths)¹² and in fluid $\delta^{30}\text{Si}$ values between -2.05‰ (soil solutions)¹³ and +4.66‰
(freshwater)¹⁴. Pacific Ocean $\delta^{30}\text{Si}$ values are mainly controlled by diatom uptake in surface
waters, subsequent dissolution and water mass mixing, spanning a range from +0.8‰ in
Northern Pacific deep water masses to +4.4‰ in the photic zone (average deep Pacific: +1.2
± 0.2‰, 1SD)^{15,16}; for a compilation see Sutton et al.¹⁰. During IODP Expedition 366, pore
fluids from serpentinite seamounts in the Mariana and Izu-Bonin arc region were sampled, in
order to study Si isotope fractionation during serpentinite-seawater alteration reactions.
Additionally, radiogenic Sr and stable B isotopes ($\delta^{11}\text{B}$) were investigated to further unravel
fluid sources and fractionation mechanisms. Serpentine can incorporate large amounts of B
in its crystal structure (up to 100 $\mu\text{g g}^{-1}$) and preferentially incorporates the ¹⁰B isotope,
enriching associated fluids with the ¹¹B isotope¹⁷⁻²⁰. Despite this distinct fractionation
behavior, a large range of $\delta^{11}\text{B}$ values in serpentinites has been detected to date, ranging
from +5.4-15.3 to +40.7‰¹⁷⁻²². B isotopes have been used to study mineral reactions as B is
an incompatible element that ~~strongly~~ partitions into the liquid phase ($D_{(\text{serp}/\text{H}_2\text{O})} = 0.25$ ²³)
and is therefore a useful tracer to identify fluid-rock processes²⁴, including serpentine
alteration. Coupling of Si and B isotopes should result in a positive correlation, in that the
light isotope is incorporated in the authigenic mineral for both isotope systems. The
combination of a new tracer (Si) with a well-established tracer (B) will help to reveal
processes during seawater alteration of serpentine and authigenic mineral precipitation. The

isotopic results were further evaluated by transport-reaction modelling to quantify rates of serpentine alteration reactions in contact with seawater and isotopic fractionation.

1. Results and discussion

The Mariana and Izu-Bonin forearc is the only region on Earth where serpentinite seamounts form above a non-accretionary convergent plate margin. Permeable faults serve as long-lived pathways for deep-sourced fluids and serpentinized mud to ascend to the surface²⁵⁻²⁷. Two seamounts were investigated (Yinazao and Fantangisña), which are located at 55 km and 62 km from the trench axis, respectively (Fig. 1a, b). Three drill cores were recovered from Yinazao (1491B and C, 1492B) and three from Fantangisña seamount (1498A and B, 1497B; Fig. 1a, b; Table S1). Drill cores were taken from the flanks of the seamounts (except coring location 1497B, which was located close to the seamount summit), in order to study mineral reactions during water-rock reactions induced by seawater circulating through the mount flanks. Thermal gradients induce temperature changes of $\leq 2^\circ\text{C}$ from the shallowest (about 4°C) to the deepest sample (about 6°C) for both seamounts²⁷. Recovered material in the uppermost 4 - 5 m core sections (~~about < 5 m depth~~) consisted of pelagic clays at Yinazao and sandy silt at Fantangisña seamount. Deeper parts of the seamounts were composed of serpentinite mud which contained xenoliths of the underlying forearc crust and mantle, and the subducting plate, ~~and the forearc mantle~~²⁷. A detailed description of the lithostratigraphic units of ~~all the investigated~~ seamounts can be found in Fryer et al.²⁷.

~~Fluids from both seamounts showed increasing~~ The Si concentrations in fluids from both seamounts vary unsystematically with ~~sediment~~ depth (range from 142-44 to 516 μM Si, Table S1; NW Pacific seawater: 146 μM Si¹⁵). ~~Along with increasing~~ In contrast to the Si concentrations, Si isotope values generally also increased from $+0.4\text{‰}$ to $+5.2\text{‰}$ with depth, with the majority being higher compared to deep NW Pacific seawater ($\delta^{30}\text{Si}_{\text{SW}}$: $+1.05\text{‰}$ ¹⁵). The maximum $\delta^{30}\text{Si}$ value in the fluids sampled at the mount flanks ($+5.2 \pm 0.2\text{‰}$; 2SD_{external reproducibility}) constitutes the highest natural fluid $\delta^{30}\text{Si}$ measured to date (Fig. 1c). In contrast to the large isotope variability observed in the fluids, serpentinite muds show only a small range in $\delta^{30}\text{Si}$ between -0.6‰ and $+0.1\text{‰}$, independent of seamount and sampling site (Table S2, Fig. S1). The fluids are further characterized by high B concentrations (from 245 to

97 758 μM B) with a wide range in $\delta^{11}\text{B}$ (+16.1 to +43.5‰), which encompasses $\delta^{11}\text{B}$ of
98 seawater (seawater B concentration: 432.6 μM ; $\delta^{11}\text{B}_{\text{SW}}$: +39.6‰²⁸) (Fig. 1c). ~~In contrast to~~
99 ~~the large isotope variability observed in the fluids, serpentinite muds show only a small~~
100 ~~range in $\delta^{30}\text{Si}$ between -0.6‰ and +0.1‰, independent of seamount and sampling site~~
101 ~~(Table S2, Fig. S1). A small shift to higher $\delta^{30}\text{Si}$ values with depth is present, however, the~~
102 ~~detected range is within measurement uncertainty and therefore to be interpreted with~~
103 ~~caution.~~

104 The combination of pore fluid $\delta^{11}\text{B}$ and $\delta^{30}\text{Si}$ values shows a very good correlation of these
105 two isotope systems (Fig. 2). For both systems, the isotope values are below and above the
106 seawater signature, indicating dissolution of presumably serpentine and precipitation of
107 authigenic mineral phases, respectively. In order to decipher the processes controlling
mineral dissolution and precipitation, fluid sources and compositional changes in relation to
depth need to be examined.

*1.1 Origin of pore fluids*

Two major fluid sources can be distinguished based on magnesium (Mg^{2+}), strontium (Sr),
and chloride (Cl^-) concentrations, Radiogenic-radiogenic Sr isotopes, and pH (Fig. 3, 4a-c).
~~can be used to identify fluid sources due to their distinct endmember signatures for deep-~~
~~sourced mantle fluids ($^{87}\text{Sr}/^{86}\text{Sr}$ of 0.70535: ODP Site 1200 at South Chamorro Seamount²²)~~
~~and modern seawater ($^{87}\text{Sr}/^{86}\text{Sr} = 0.7092^{27}$).~~ Most of the pore fluids show seawater-like
$^{87}\text{Sr}/^{86}\text{Sr}$ signatures in the upper ~30 m (surface layer), except samples from sites ~~1491B,~~
1498A and B (Fig. 3, 4a-c). Samples from site 1498B overlap with the local fluid mantle
endmember and can thus be identified as mantle fluids, showing depleted Mg^{2+}
concentrations, high pH values (~11), decreasing Cl^- concentrations, and increasing Sr
concentrations, characteristic trends reported for deep mantle fluids from the Mariana
forearc region^{25,29}. Also, a $^{87}\text{Sr}/^{86}\text{Sr}$ ratio similar to deep-sourced mantle fluids is measured in
these fluids ($^{87}\text{Sr}/^{86}\text{Sr}$ of 0.70535: ODP Site 1200 at South Chamorro Seamount²⁵). (see also
lower red circle in Fig. 1b). Therefore, we interpret the related low $\delta^{30}\text{Si}$ and $\delta^{11}\text{B}$ isotope
values (+1.6‰ and +16.1‰, respectively) to originate from a representative isotope
signatures for the deep mantle fluids affected by during pervasive serpentinization. Sample
1498A 15-R-1 shows a high Sr concentration (189 μM) and a less-more radiogenic $^{87}\text{Sr}/^{86}\text{Sr}$

signature (0.70763; Fig. 4) compared to the mantle-derived fluids and for sample 1498A likely results from mixing with the local fluid mantle endmember and Sr released from mafic clasts during reaction with between the two endmembers (seawater and deep-mantle fluid) discussed above seawater²⁸.

1.2 Identification of fractionation processes

1.2.1 Drivers of deep mantle-derived $\delta^{30}\text{Si}$ and $\delta^{11}\text{B}$ values

During serpentinization, pore fluid pH can increase rapidly to alkaline values (pH ~9 - 12²⁷), so that dissolved B is tetrahedrally coordinated. This is also the species preferentially bound in the serpentine mineral structure, and so limited fractionation between mineral and water is expected taking also the high formation temperatures into account (average T of 200°C)^{19,30}. The mantle-derived pore fluids of site 1498B have the lowest $\delta^{11}\text{B}$ values of about +16‰ and the lowest B concentrations (between 84 and 111 μM ; Table S1). These low $\delta^{11}\text{B}$ values overlap within error with serpentinized peridotite clasts and serpentinite matrix identified by Benton et al.¹⁷ with $\delta^{11}\text{B}$ values of about +14‰ for Conical seamount in the Mariana forearc. During serpentinization, the pH increases rapidly to alkaline values (pH ~9-12), so that B in solution is tetrahedrally coordinated, which is also the species preferentially build in the serpentine mineral structure. Consequently, we conclude that no significant B isotope fractionation occurs during early pervasive high pH serpentinization reactions. In contrast, Si isotopes vary between pore-potentially mantle-derived fluids ($\delta^{30}\text{Si} = +1.6\text{‰}$) and serpentinite muds (on average $\delta^{30}\text{Si} = -0.3 \pm 0.2\text{‰}$; 1SD; Table S2). However, by comparing pristine mantle rocks ($\delta^{30}\text{Si} = -0.29 \pm 0.08\text{‰}$)³¹ with the serpentinized muds investigated in this study, we show no, or only minor, Si fractionation occurs during the transformation of olivine/ pyroxene to serpentine, as the $\delta^{30}\text{Si}$ values of the serpentine muds overlap with the $\delta^{30}\text{Si}$ of mantle rocks ($-0.29 \pm 0.08\text{‰}$)³¹ and This also confirms the isochemical nature of pervasive serpentinization³. The pore fluid $\delta^{30}\text{Si}$ value is likely affected by Si isotope fractionation during ascent and deposition of the serpentinite muds and accompanied cooling, which induces Si precipitation and fractionation of Si isotopes on the slope of the seamounts, where serpentine dissolution and Si re-precipitation occurs (see also section 1.2.3). Interestingly, the expected Si isotope fractionation dependence on pH (the dominant

species H_4SiO_4 at $\text{pH} < 8.5$ dissociates to H_3SiO_4^- and $\text{H}_2\text{SiO}_4^{2-}$ at $\text{pH} > 8.5$, whereby the ^{28}Si
isotope fractionates preferentially in the dissociated Si species³²⁻³⁴ appears to have little to
no effect during serpentine formation as recorded here.

*1.2.2 Shallow low $\delta^{30}\text{Si}$ and $\delta^{11}\text{B}$ values: clay dissolution versus organic matter* 166 *degradation*

Pore fluids with low $\delta^{30}\text{Si}$ and $\delta^{11}\text{B}$ values compared to seawater originate from the shallow,
uppermost part of the seamounts (depth < 5 m depth; except samples from 1498B, which
originate from the mantle, see also section 1.1), which is dominantly composed of pelagic
clay and silt²⁶. Pelagic clay reacting with seawater releases adsorbed B and thus increases B
concentrations³⁵. The borate ion $[\text{B}(\text{OH})_4^-]$ is the preferred B species which adsorbs to clay
minerals and with it the ^{10}B , which is the isotope preferentially fractionated into this species.
During desorption in reaction with seawater, the ^{10}B is thus re-released into the pore fluid
shifting $\delta^{11}\text{B}$ to lower values compared to seawater. In general, organic matter
decomposition might also contribute to this isotopic shift as also the ^{10}B isotope was found
to be preferentially released during this process^{35,36}. However, as alkalinity, phosphate, and
ammonia are not reported to do not increase (Table T6 in Fryer et al.²⁶), organic matter
decomposition can be excluded and pelagic clay dissolution is the most likely process. As
clays preferentially incorporate the ^{28}Si isotope³⁷, the process of pelagic clay dissolution at
shallow depths can also explain the low $\delta^{30}\text{Si}$ values ($\delta^{30}\text{Si}$ from +0.4‰ to +0.8‰; Fig. 1, 3a).

*1.2.3-2 Shallow high $\delta^{30}\text{Si}$ and $\delta^{11}\text{B}$ values: processes of serpentine alteration*

The investigation of Si and B isotopes revealed a similar fractionation response during
serpentine alteration and shows the potential of coupled Si and B isotope data to trace
serpentine alteration reactions (Fig. 2). Interestingly, despite generally increasing Si and B
concentrations, the isotope values increase as well. This observation seems counterintuitive,
in that concentration increases generally relate to mineral dissolution processes, which are
not associated with significant isotopic fractionation³². Motivated by this observation, a one-
dimensional transport-reaction model (see Methods and supplementary materials for model
details) was set up to constrain– for the first time– the seawater entrainment rate, the

precipitation and dissolution rates, and the isotopic fractionation during authigenic mineral
formation. This model is a simplification as it does not capture the additional changes which
are likely induced by the 3-D geometry of the seamounts, and the time- and space-
dependency of mixing rates and fluid flow velocities.

The depth profiles indicate a seamount surface layer with a thickness of about 30 m where
bottom water is entrained into the muds by shallow seawater circulation and an underlying
layer that is affected by the ascent of deep fluids originating from the subducted slab (Fig. 3,
4). The strong increase in Si and B concentrations in the surface layer is induced by rapid
dissolution of serpentine, which is undersaturated with respect to serpentine due to the
entrainment of pH-neutral seawater (Fig. 5). The persistent mixing in the surface layer with
seawater supports high dissolution rates ($\sim 0.9 \mu\text{M cm}^{-3} \text{ yr}^{-1}$) whereas the rates are low in the
deeper layers ($< 10^{-5} \mu\text{M cm}^{-3} \text{ yr}^{-1}$) that are not affected by mixing with ambient seawater
(Fig. 5). Serpentine, which formed in the mantle wedge, becomes unstable after deposition
on the seafloor either as serpentinitized clasts or as serpentinite mud and begins to dissolve
during interaction with seawater via hydrolysis³³, following:

thereby releasing Si and B into the pore fluid (see also upper red circle in Fig. 1b; Fig. 6a,b).
The $\delta^{30}\text{Si}$ and $\delta^{11}\text{B}$ values increase in the about upper 30 m of the seamounts even though
isotopically depleted Si ($\delta^{30}\text{Si} = -0.33\text{‰}$; Table S2) and B ($\delta^{11}\text{B} = +16\text{‰}$; see section 1.2.1) are
released by the dissolving serpentine. This observation can only be explained by formation
of authigenic minerals removing isotopically depleted Si and B from the pore fluids (Fig. 2, 4).
The formation of authigenic minerals was simulated by allowing talc ($\text{Mg}_3\text{Si}_4\text{O}_{10}(\text{OH})_2$)
precipitation in the model. Talc was chosen because thermodynamic equilibrium calculations
conducted with PHREEQC³⁴ show that the surface layer of the seamounts is strongly
oversaturated with respect to this mineral (Fig. 5). Other Al-containing silicate phases may
form as well; however, we were not able to simulate the formation of these phases due to
the lack of dissolved Al data. The modeling indicates that a large fraction of the Si and B
released from the dissolving serpentine is removed from solution by authigenic mineral
precipitation (Fig. 5). We obtained a good fit to the $\delta^{30}\text{Si}$ data applying a Si isotope
fractionation of $\Delta^{30}\text{Si} = -3\text{‰}$, where $\Delta^{30}\text{Si}$ is defined as $\Delta^{30}\text{Si} = \delta^{30}\text{Si}_{\text{mineral}} - \delta^{30}\text{Si}_{\text{pore fluid}}$. Lower

and higher Si isotope fractionation could not reproduce the observed $\delta^{30}\text{Si}$ values of the
sample as tested by model runs using $\Delta^{30}\text{Si}$ of -2‰ and -4‰ (Fig. 6c). This Si isotope
fractionation is the highest abiotic fractionation observed for authigenic mineral formation
so far. In natural systems, only Si precipitation from supersaturated solutions to form
amorphous silica and experimentally investigated Si adsorption to Al shows $\Delta^{30}\text{Si}$ values in
this range or higher (up to -5‰)^{35–37}.
The almost complete removal of dissolved Si by authigenic mineral precipitation induces a
strong $\delta^{30}\text{Si}$ maximum at the base of the surface layer, which is the highest pore fluid $\delta^{30}\text{Si}$
value that has been observed in natural environments. There is only a limited data set
available on marine pore fluid $\delta^{30}\text{Si}$, mostly originating from continental margin settings. The
$\delta^{30}\text{Si}$ values range between -0.5 to +2.5‰ and are dominantly controlled by dissolution of
biogenic silica and the formation of authigenic clays with a $\Delta^{30}\text{Si}$ of -2‰^{38–40}.
The Si isotope fractionation depends on the chemical composition and crystal structure of the authigenic
mineral, the precipitation rate, temperature, and successive mineral precipitation^{35,36,41,42}.
We can only speculate that a combination of slow reaction rates associated with low
temperatures (authigenic mineral precipitation rate $\sim 0.8 \mu\text{M Si cm}^{-3} \text{ yr}^{-1}$ at $\sim 4^\circ\text{C}$; Fig. 5) and a
241 Mg-rich authigenic mineral composition drives the Si isotope fractionation to higher values
at the Mariana seamounts compared to continental margin settings (e.g. Peruvian margin)
with faster reaction rates at higher temperatures (authigenic mineral precipitation rate up to
$\sim 27 \mu\text{M Si cm}^{-3} \text{ yr}^{-1}$ at $\sim 11^\circ\text{C}$; Ehlert et al.³⁸) and an Al-rich authigenic mineral composition.
This assumption needs to be constrained in further studies, however, similar reactions and
associated Si isotope fractionation as observed for the Mariana seamounts may occur in
other settings, where oceanic crust and serpentinite interact with seawater at low
temperatures (see section 1.3).
The isotopic fractionation of B during mineral precipitation is related to two processes: the
pH-dependent fractionation between borate and boric acid in the pore fluid and the
fractionation during borate uptake in the solid phase³⁰.
Hence, we calculated the concentrations of borate and boric acid and their isotopic composition applying the
dissociation constant for boric acid in seawater⁴³ and the isotopic equilibrium constant for
borate and boric acid⁴⁴. The model results confirm that dissolved borate has a much lower
$\delta^{11}\text{B}$ than total dissolved boron in the surface layer (Fig. 6d) such that ^{10}B is removed from

solution when borate is preferentially bound in the authigenic silicate phase. The modeling
showed that an additional fractionation between dissolved borate and solid phase borate
has to be applied to match the $\delta^{11}\text{B}$ data. The best fit was attained applying an isotopic
fractionation factor of $\Delta^{11}\text{B}_{\text{mineral-borate}}$ of -20‰ where $\Delta^{11}\text{B}_{\text{mineral-borate}}$ is defined as $\delta^{11}\text{B}_{\text{borate}}$
$\text{authigenic mineral} - \delta^{11}\text{B}_{\text{dissolved borate}}$. This B_{borate} isotope fractionation results in an average $\delta^{11}\text{B}$
value of the authigenic clay of $-2.4 \pm 2.0\%$ (1 SD), supposing an average of the modeled
$\delta^{11}\text{B}_{\text{borate}}$ in the upper 15 m (highest talc precipitation rate) of $+17.6\%$. This results in a
general B isotope fractionation between authigenic mineral and the pore fluid
of $-43.3 \pm 2.3\%$ (1 SD), with $\Delta^{11}\text{B}_{\text{mineral-pore fluid}} = \delta^{11}\text{B}_{\text{mineral}} - \delta^{11}\text{B}_{\text{pore fluid}}$ (average of the modeled
$\delta^{11}\text{B}_{\text{pore fluid}}$ in the upper 15 m of $+40.9\%$). This B fractionation fits well to modeled values for
low temperature phyllosilicate formation by Boschi et al.¹⁸ based on data by Liu & Tosse³⁰.
The coupling of Si and B isotopes showed that the light isotope is preferentially incorporated
in authigenic minerals and that the combination of both isotope systems can trace
serpentine alteration reactions (Fig. 2, 6). Pore fluids with $\delta^{30}\text{Si}$ and $\delta^{11}\text{B}$ values lower than
seawater ($\delta^{30}\text{Si}$ from $+0.4\%$ to $+0.8\%$; $\delta^{11}\text{B}$ from $+37.5\%$ to $+38.2\%$; Fig. 1, 2) originate
from the shallow, uppermost part of the seamounts (depth < 5 m depth; except samples
from 1498B, which originate from the mantle, see also section 1.1). A possible reason for
these low isotopic values might be the dissolution of authigenic minerals, which formed
during potentially higher upward fluid flow velocities in the past. Lower fluid flow velocities
in the present would cause dissolution of these authigenic phases as seawater is the
dominant ambient fluid phase. Fluid flow velocities can vary significantly between the
seamounts in the Mariana forearc region (ie between 2 mm yr^{-1} to 4 cm yr^{-1})²⁵ and thus
changes over time are not unlikely as well. More data is needed to unequivocally disentangle
additional process occurring during serpentine alteration, however, our data show for the
first time the potential of coupled Si and B isotope data to trace serpentine alteration
reactions.

285 *1.3 Impact of seafloor alteration on the global Si (isotope) cycle*

The findings of this study have important implications for the global oceanic Si cycle and
isotope budget, as seafloor alteration of serpentinites is found to be a source of Si with

extremely high $\delta^{30}\text{Si}$ values. So far, none of the global marine Si models or marine budget
calculations have considered low temperature serpentine alteration, which might be a
common process as serpentinites are widespread on the seafloor especially in the vicinity of
Mid Ocean Ridges (MOR), transform and bend faults, and subduction zones⁵. In general,
alteration of serpentinites by circulating seawater has received little attention, even though
previous studies have shown that low temperature weathering of abyssal serpentinites leads
to significant modifications in mineralogy, chemical composition, and physical
properties indicating^{45,46} Serpentinites may contribute up to 20 vol.% of the oceanic
basement close to MORs and up to 16.7 vol.% in the vicinity of subduction zones ~~(low~~
~~temperature alteration at the seafloor)~~^{6,7}. Serpentine alteration results in the ~~After initial~~
~~high temperature serpentine formation, low temperature alteration on the seafloor results~~
~~in the~~ formation of authigenic secondary minerals like aragonite, Fe-oxyhydroxides, and
clays, for example in the abyssal serpentinites at the Iberia abyssal plain³³, the Puerto Rico
Trench⁴⁵, and from the Mid Atlantic Ridge⁴⁷. The clay mineral assemblages are identified as
303 Mg-Fe-smectite, with saponite and montmorillonite as dominating phases^{33,48}. Saponites as
well as other authigenic clay minerals are reported as authigenic minerals in Yinazao and
Fantangisña seamounts⁴⁹ and resemble in their mineralogy (Mg-rich silicates) closely talc,
the mineral phase used in the model approach (see section 1.2.2). Therefore, we propose
that geochemical results from the altered Mariana forearc serpentinites can serve as an
example for alteration of oceanic serpentinites in general, implying that ~~This secondary~~
~~mineral assemblage is similar to that observed within Mariana seamounts and is thus likely a~~
~~global phenomenon. This study shows that associated with these low temperature~~
~~alteration reactions~~ of marine serpentinites result in ~~are~~ elevated Si concentrations and high
$\delta^{30}\text{Si}$ values of reacting fluids. Further, we hypothesize, that serpentine alteration reactions
are directly comparable to low temperature, off-axis basalt alteration, given that the most
abundant authigenic mineral replacing igneous phases is also saponite⁵⁰. Serpentine
~~alteration reactions are comparable to basalt alteration occurring at the seafloor, induced by~~
~~off-axis or ridge flank circulation. At ridge flanks, fluid discharge is hampered by overlying~~
~~sediments, which act as low permeability barrier and instead, the fluid is channeled through~~
~~permeable features, like seamounts, faults or abyssal hills. Discharged fluids are enriched in~~
~~Si and are controlled by reaction temperatures in the basement and precipitation of~~
~~secondary clays. Associated net~~ Effusing fluids associated with off-axis basalt alteration have

Si concentrations ~~range~~ between 200-550 $\mu\text{M Si}$ ^{51,52}, which is identical to the range observed
in this study. ~~Basalt alteration during ridge flank circulation may release 0.4 Tmol Si yr⁻¹~~
~~making up 3% of the riverine influx, the largest Si source to the ocean, not yet taking into~~
~~account Si input from low-T serpentine alteration.~~ According to our new data the associated
Si fluid isotope signature with serpentine and likely basalt alteration ~~is~~ may be high (on
average +32.7±0.3‰, locally even up to +5.2‰; Table S1). We further hypothesize, that the
Si isotope signature of the oceanic crust shifts away from the basaltic endmember to lower
values with increasing open-system alteration states due to increasing formation of
authigenic clays. This process might additionally be enhanced by an increasing degree of
mineral crystallization with crustal age⁵⁰ and an associated lower clay $\delta^{30}\text{Si}$ value⁵³. The
impact on oceanic $\delta^{30}\text{Si}$ values needs to be assessed by constraining the Si fluxes and
associated $\delta^{30}\text{Si}$ values related to serpentine and basalt alteration and authigenic mineral
formation. Oceanic crust alteration may also influences oceanic $\delta^{30}\text{Si}$ values at regional scale,
especially in restricted basins and may potentially explain, at least in part, Si anomalies ~~in the~~
~~open ocean such as~~ the Northeast Pacific Silicic Acid Plume (NPSP). The NPSP has unusually
high Si concentrations ($> 150 \mu\text{M Si}$) and high $\delta^{30}\text{Si}$ signatures of +1.5‰, the highest Pacific
deep ocean Si isotope value^{54,55} (Fig. 67), between 2000 and 3000 m water depths,
originating mostly from the Cascadia Basin (about 200 $\mu\text{M Si}$)⁵⁶. This has not been en explained
unambiguously to date. Findings of this study show that low-temperature alteration of
serpentinites and potentially seafloor basalts are associated with high Si concentration and
$\delta^{30}\text{Si}$ values which are expelled to the ocean. ~~Such~~ a source is likely to contribute to the
unusually high NPSP and related Si isotope values (see Supplement for details). Low-
temperature seafloor may contribute significantly to the $\delta^{30}\text{Si}$ budget of the ocean ~~alteration~~
~~is likely an important Si source to the global marine Si cycle, considering given~~ the large areas
of serpentinitized oceanic crust and the related fluid discharge with extremely high Si isotope
values.

Acknowledgements

We appreciate the support of the master and crew of the JOIDES Resolution during the IODP
Expedition 366 Science Party. We thank Regina Surberg and Ana Kolevica for analytical support.
Further thanks go to Jutta Heinze, Daphne Bartels, and Tyler Goepfert. IODP Germany and ECORD are

thanked for financial support as well as IODP grant code: NERC UK IODP Phase 2 Moratorium Award
NE/P020909/1.

**Author contributions**

CM was involved in research cruise IODP366 and carried out the sampling. SG performed the Si and B
isotope analyses. VL conducted the Sr isotope measurements. KW ~~supported the model-~~
~~based~~ designed the transport-reaction model-evaluation-of isotope values. SG, PG, CM, and KW
interpreted the isotope data. SG wrote the manuscript with help from PG, CM, VL, and KW.

[revised manuscript text omitted]

- 49. Morrow, C. A., Moore, D. E., Lockner, D. A. & Bekins, B. A. Data report: permeability,
porosity, and frictional strength of core samples from IODP Expedition 366 in the
Mariana forearc. in Fryer, P., Wheat, C.G., Williams, T., and the Expedition 366
Scientists, Mariana Convergent Margin and South Chamorro Seamount. *Proceedings*
*of the International Ocean Discovery Program, 366: College Station, TX (International*
*Ocean Discovery Program)* (2019).
- 50. Coogan, L. A. & Gillis, K. M. Low-Temperature Alteration of the Seafloor: Impacts on
Ocean Chemistry. *Annu. Rev. Earth Planet. Sci.* **46**, 21–45 (2018).

- 51. Wheat, C. G., Fisher, A. T., McManus, J., Hulme, S. M. & Orcutt, B. N. Cool seafloor
hydrothermal springs reveal global geochemical fluxes. *Earth Planet. Sci. Lett.* **476**,
179–188 (2017).
- 52. Wheat, C. G. & McManus, J. The potential role of ridge-flank hydrothermal systems on
oceanic germanium and silicon balances. *Geochim. Cosmochim. Acta* **69**, 2021–2029
(2005).
- 53. Opfergelt, S. *et al.* Iron and silicon isotope behaviour accompanying weathering in
Icelandic soils, and the implications for iron export from peatlands. *Geochim.*
*Cosmochim. Acta* **217**, 273–291 (2017).
- 54. Beucher, C. P., Brzezinski, M. A. & Jones, J. L. Sources and biological fractionation of
Silicon isotopes in the Eastern Equatorial Pacific. *Geochim. Cosmochim. Acta* **72**, 3063–
3073 (2008).
- 55. Hendry, K. R. & Brzezinski, M. A. Using silicon isotopes to understand the role of the
Southern Ocean in modern and ancient biogeochemistry and climate. *Quat. Sci. Rev.*
**89**, 13–26 (2014).
- 56. Johnson, H. P., Hautala, S. L., Bjorklund, T. A. & Zarnetske, M. R. Quantifying the North
Pacific silica plume. *Geochemistry Geophys. Geosystems* **7**, (2006).

Fig. 1. (a) Sampling area of the two seamounts (Yinazao and Fantangisña) investigated during
 IODP Expedition 366 and close up of the sample locations for both seamounts. (b) Sketch of
 the Mariana forearc region with indicated sampling locations (modified after Fryer et al.²⁷).
 Blue dotted lines indicate fluid flow along faults. Note mantle-derived fluid flow for site
 1498B (see section 1.1 and 1.2.1). (c) Isotope compositions versus depths (m) are displayed
 for $\delta^{30}\text{Si}$ and $\delta^{11}\text{B}$. Error bars are within symbol size. Brown area denotes depth of
 uppermost pelagic clays (≤ 5 m depth) and green area shows mineralogy dominated by
 serpentinites and mafic xenoliths.

Fig. 2. Pore fluid $\delta^{11}\text{B}$ versus $\delta^{30}\text{Si}$ values reveal different stages of silicate dissolution and precipitation (see text for explanation). Error bars (2SD) not indicated are within symbol size.

Fig. 23. Depth-profiles characterizing the layering of the seamount fluid system. a) Mg^{2+} concentrations in pore fluids, b) pH values, c) sediment porosity. Functions were fitted through the data to generate continuous depth profiles for the transport-reaction modeling (supplementary information). Mg^{2+} and porosity data are reported in Fryer et al. ²⁷.

Fig. 4. Depth profiles of conservative tracers. a) Dissolved Sr, b) $^{87}\text{Sr}/^{86}\text{Sr}$ ratios in pore fluids,
 c) Dissolved Cl^- (taken from Fryer et al.²⁷), d) The non-local mixing coefficient α was derived
 by fitting the model to the data (a-c). Dissolved Cl^- , Sr, and $^{87}\text{Sr}/^{86}\text{Sr}$ ratio in the pore fluid
 samples were employed to constrain the steady-state mixing rate and fluid flow velocity
 assuming that the depth-distribution of these dissolved tracers is governed by transport
 processes rather than precipitation/dissolution reactions. The best fit to these data was
 obtained applying mixing in the upper 30 m with a non-locale mixing coefficient of 5×10^{-4}
 574 yr^{-1} and an upward fluid flow velocity of 0.01 cm yr^{-1} . Thick lines indicate steady-state model
 results (supplementary information).

Fig. 5. Reaction rates applied in the modeling to obtain the model fit shown in Fig. 6. a)

Saturation state of the pore fluid with respect to serpentine, b) saturation state of the pore

fluid with respect talc. Saturation states were calculated applying measured pH values and

Mg²⁺ concentrations (Fig. 3) and modeled silica concentrations (Fig. 6). c) Rate of serpentine

dissolution, d) rate of talc precipitation. Note the different depth scale in (c) and (d).

Fig. 6. Depth profiles of reactive tracers. a) Dissolved Si, b) total dissolved B, dissolved borate

(B(OH)₄⁻) and dissolved boric acid (B(OH)₃), c)- δ³⁰Si of dissolved silica. Model data based on a

Si isotope fractionation with Δ³⁰Si of -3‰ (best fit) and sensitivity tests assuming -2‰

and -4‰ (dotted lines). d). δ¹¹B values of total dissolved boron, dissolved borate and

dissolved boric acid following Zeebe & Wolf-Gladrow⁴³ and Klochko et al.⁴⁴ (see section 1.2.2

and Supplement for details). Error bars within symbol size. Thick or labelled lines indicate

steady-state model results (supplementary information).

| Fig. 47. Mariana seamounts median fluid $\delta^{30}\text{Si}$ value (error bar equals the coefficient of
 | quartile deviation) versus the inverse Si concentration ($1/\text{Si}$ in μM^{-1}). Additionally global
 | seawater and the Cascadia Basin $\delta^{30}\text{Si}$ values are shown (modified after Beucher et al.⁵⁴ and
 | Hendry & Brzezinski⁵⁵ and references therein).

**Method**

[revised manuscript text omitted]
 \cdot \left(D_s \cdot \frac{\partial C}{\partial x} + v \cdot C \right) \right) + \Phi \cdot \alpha \cdot (C_{BW} - C) + \Phi \cdot R$$

with Φ : porosity, C : concentration of dissolved species in the pore fluid ($\mu\text{mol cm}^{-3}$), t : time
(yr), x : sediment depth (cm), D_s : molecular diffusion coefficient of dissolved species in
sediment pore fluid ($\text{cm}^2 \text{ yr}^{-1}$), v : upward fluid flow velocity of pore fluid (cm yr^{-1}), α : mixing
coefficient (yr^{-1}), R : turnover rates of dissolved species ($\mu\text{mol cm}^{-3} \text{ yr}^{-1}$). The model was set
up for the following dissolved species: Si, ^{30}Si , B, ^{11}B , Sr, ^{87}Sr , Cl. The isotopic compositions of
the pore fluids ($\delta^{30}\text{Si}$, $\delta^{11}\text{B}$, $^{87}\text{Sr}/^{86}\text{Sr}$) are calculated from the corresponding mole fractions
($^{30}\text{Si}/\text{Si}$, $^{11}\text{B}/\text{B}$, $^{87}\text{Sr}/\text{Sr}$) applying previously published approaches^{16,17} and the boundary
conditions, equations and parameter values given in the supplementary materials.

**Method References**

- 1. Fryer, P., Wheat, C. G., Williams, T. & Expedition 366 Scientists, T. Mariana
Convergent Margin and South Chamorro Seamount. *Proc. Int. Ocean Discov.*
*Program, 366 Coll. Station. TX (International Ocean Discov. Program) (2018).*
- 2. Gieskes, J. M., Gamo, T. & Brumsack, H. Chemical methods for interstitial water
analysis aboard Joides Resolution. *Ocean Drill. Prog. Tech. Note 15. Texas A&M Univ.*
*Coll. Stn. (1991).*
- 3. van den Boorn, S. H. J. M. *et al.* Determination of silicon isotope ratios in silicate
materials by high-resolution MC-ICP-MS using a sodium hydroxide sample digestion
method. *J. Anal. At. Spectrom.* **21**, 734 (2006).
- 4. Georg, R. B., Reynolds, B. C., Frank, M. & Halliday, A. N. New sample preparation
techniques for the determination of Si isotopic compositions using MC-ICPMS. *Chem.*
*Geol.* **235**, 95–104 (2006).
- 5. van den Boorn, S. H. J. M., Vroon, P. Z. & van Bergen, M. J. Sulfur-induced offsets in
MC-ICP-MS silicon-isotope measurements. *J. Anal. At. Spectrom.* **24**, 1111 (2009).
- 6. Hughes, H. J. *et al.* Controlling the mass bias introduced by anionic and organic
matrices in silicon isotopic measurements by MC-ICP-MS. *J. Anal. At. Spectrom.* **26**,
1892 (2011).
- 7. Ehlert, C. *et al.* Stable silicon isotope signatures of marine pore waters – Biogenic opal
dissolution versus authigenic clay mineral formation. *Geochim. Cosmochim. Acta* **191**,
102–117 (2016).
- 8. Cardinal, D., Alleman, L. Y., de Jong, J., Ziegler, K. & André, L. Isotopic composition of
silicon measured by multicollector plasma source mass spectrometry in dry plasma
mode. *J. Anal. At. Spectrom.* **18**, 213–218 (2003).
- 9. Oelze, M., Schuessler, J. A. & von Blanckenburg, F. Mass bias stabilization by Mg
doping for Si stable isotope analysis by MC-ICP-MS. *J. Anal. At. Spectrom.* **31**, 2094–
2100 (2016).
- 10. Albarède, F. *et al.* Precise and accurate isotopic measurements using multiple-
collector ICPMS. *Geochim. Cosmochim. Acta* **68**, 2725–2744 (2004).
- 11. Grasse, P. *et al.* GEOTRACES inter-calibration of the stable silicon isotope composition
of dissolved silicic acid in seawater. *J. Anal. At. Spectrom.* **32**, 562–578 (2017).
- 12. Gaillardet, J., Lemarchand, D., Göpel, C. & Manhès, G. Evaporation and sublimation of
boric acid: Application for boron purification from organic rich solutions. *Geostand.*
*Newsl.* **25**, 67–75 (2001).
- 13. Jurikova, H. *et al.* Boron isotope systematics of cultured brachiopods : Response to
acidification , vital effects and implications for palaeo-pH reconstruction. *Geochemica*
*Cosmochinica Acta* **248**, 370–386 (2019).
- 14. Vogl, J. & Rosner, M. Production and Certification of a Unique Set of Isotope and
Delta Reference Materials for Boron Isotope Determination in Geochemical,
Environmental and Industrial Materials. *Geostand. Geoanalytical Res.* **36**, 161–175
(2012).
- 15. Howarth, R. J. & McArthur, J. M. Strontium isotope stratigraphy. in *A Geological Time*
*Scale, with Look-up Table Version 4* (eds. Gradstein, F. M. & Ogg, J. G.) 96–105
(Cambridge University Press, Cambridge, U.K pp. ., 2004).
- 16. Ehlert, C. *et al.* Stable silicon isotope signatures of marine pore waters – Biogenic opal
dissolution versus authigenic clay mineral formation. *Geochim. Cosmochim. Acta* **191**,
(2016).

- 17. Geilert, S. *et al.* Impact of ambient conditions on the Si isotope fractionation in
marine pore fluids during early diagenesis. *Biogeosciences* **17**, 1745–1763 (2020).

REVIEWER COMMENTS

Reviewer #1 (Remarks to the Author):

Re-review of manuscript NCOMMS-19-35155 submitted to Nature Communications by Sonja Geilert and colleagues: Serpentine alteration and the impact on the marine Si cycle

With apologies that this review is slightly overdue.

Geilert and colleagues present a manuscript that details silicon, boron, and strontium isotope geochemistry ($\delta^{30}\text{Si}$, $\delta^{11}\text{B}$, $87\text{Sr}/86\text{Sr}$) of fluids and muds from seamounts experiencing serpentinization and subsequent serpentine alteration. They argue for no significant Si isotope fractionation during serpentinization reactions, but substantial fractionation during the subsequent dissolution of serpentine and (re)precipitation of authigenic mineral phases (that they represent by talc in their model). This creates fluids with $\delta^{30}\text{Si}$ locally $>5\text{‰}$, the highest fluids measured to date, and a potentially important source in the ocean Si budget. In this revised version, the manuscript benefits from a more quantitative model and clarification of several points. I think the central conclusion – that high $\delta^{30}\text{Si}$ fluids are generated by the alteration (and reprecipitation) of serpentine minerals/muds – is robust, although the complexities of natural systems will always hinder any attempts to numerically model the processes. While I have some quibbles about the RTM, I think the novelty of the data and the general interpretative framework are sufficient to warrant publication.

In my original review of the manuscript, I raised several (overlapping) points that are now largely dealt with in the revised version. The largest change is the incorporation of a reactive transport model, and this helps to address many of the issues. Specifically:

- the descriptive nature of the results: the new model goes some way to addressing this. As and aside, I would also encourage the authors to make their model code accessible. Perhaps I miss it in the online portal, but simply displaying an advection-diffusion-reaction equation (L671/SL 17) is not the same as making the underlying scripts available.

- the generalizability of the interpretations to other regions of the ocean: the authors have made a good attempt at justifying the extension of their results to elsewhere (e.g. revised ms section 1.3), while also emphasising that this would need to be demonstrated in other field studies. I find this reasonably convincing.

- the role of temperature as a potential control on fractionation: I had misinterpreted the potential range of temperature experienced by their fluids today (i.e. only with a few deg C, L78 in revised ms). However, the response does not account for the thermal history of the circulating seawater, but I interpret from e.g. Fig 4d that the penetration of seawater into the seamounts is not deep – and therefore not at elevated T, as otherwise it would have distinct Sr isotope composition etc.? Perhaps mention this around L78 for the sake of clarity, if so.

- issues with the numerical model previously used (e.g. assumptions associated with Rayleigh distillation model, etc.). These are largely resolved with the RTM model now employed, though other issues now appear. For example, model sensitivity to the prescribed parameters (Tables S3,4,5) is not investigated. Perhaps this is not necessary: my understanding is that the true value of model lies not in its ability to pinpoint one precise set of parameters, but rather to show that the dissolution of

serpentine and subsequent re-precipitation of secondary phases (whether talc, or something else) can produce fluid $\delta^{30}\text{Si}$ in the same range as the observations. One thing that I missed is a quantification of the fraction of Si being re-sequestered into the talc. Presumably there is a trade-off between fluid $\delta^{30}\text{Si}$ and fluid [Si] - as $\delta^{30}\text{Si}$ increases, [Si] decreases. Does this have implications for the importance of the flux at a regional- to global-scale?

The issues raised by other reviewers included the utility of $\delta^{11}\text{B}$, and the ability of extrapolating insights or fluxes from serpentine alteration reactions (as done here) to the broader suite of basalt alteration reactions. I think the authors make a reasonable case in the revised version as to the benefits of a dual-isotope approach, and the arguments based on secondary phase mineralogy are helpful in assessing the extension of the conclusions to other regions of the ocean. However, it seems to me that $\delta^{11}\text{B}$ is not being used as a well-understood tracer to 'calibrate' the $\delta^{30}\text{Si}$ data, as the text/rebuttal seems to imply, but rather as a system that is thought to largely be sensitive to the same processes (i.e. secondary phase formation), so that correlation between the two supports the interpretations, and aids in their model parameterisation. I think this is OK, but if this is not the case then the text should be clarified to highlight where some knowledge of the system is inferred on the basis of $\delta^{11}\text{B}$ and/or [B] alone, and then the silicon isotope interpretation made on the basis of this knowledge.

Some minor comments on the revised version of the manuscript:

L151: "Motivated".

L162: "the rapid dissolution of serpentine, which is undersaturated with respect to serpentine" – something is off here, please rephrase.

Figure 7: Not clear what the line is. Are the north Pacific data points not the most relevant here?

Are all figures necessary in the main text? It seems like 3 and 4 could potentially be combined, while 5 might be better suited in the supplement.

On figure 2, is it possible to display a) mixing hyperbolas between seawater and mantle fluid, and b) model predicted vectors of fluid evolution in $\delta^{30}\text{Si}$ - $\delta^{11}\text{B}$ space? This would allow the reader to assess the extent to which the general assumptions implicit in the RTM can replicate the range of data observed.

Reviewer #2 (Remarks to the Author):

Dear editor,

this revised manuscript has addressed most of the comments from the reviewers. I am happy with their revision. I would like to suggest acceptance if the authors can address a few more things.

1. A few words need further clarification.

line 139: change Si fractionation to Si isotope fractionation.

line 149: what is "isotope values"?

line 241: change "More data is needed" to "More data are needed".

2. line 57-58: "B is an incompatible element". "incompatible element" is not referred to the partitioning behaviour between fluid and solid, but melt and solid.

3. discussion about altered oceanic crust.

line 275-277 and line 281-283. Yu et al. (2018 Chemical Geology) published Si isotope data for altered oceanic crust. the observation should be connected with the discussion here if the authors do want to predict the Si isotope fractionation of altered oceanic crust.

all the best,

-Fang Huang

Reviewer #3 (Remarks to the Author):

General comments

All my comments from my previous review have been addressed in a satisfactory way. They provide an updated manuscript. It retains their Si and B isotope data and provide a novel approach to understand the role of serpentinites and its alteration in the Si global cycle. They added a new section using a one-dimensional transport-reaction model to explain their data and estimate fractionation factors during serpentinite formation and alteration. Dealing with open systems is always problematic and the authors have done a good work in addressing the problematics and come with a model to interpret their data that is solid. Overall their model requires significant serpentine dissolution and loss of some Si from the serpentine muds to seawater. This is shown using pore fluid $d_{30}\text{Si}$ and I think is a solid result.

However, before I recommend this paper for publication it requires some minor changes in the text as well as some important clarifications with data treatment and figure 2.

Specific comments

Line 30 I suggest that the authors change seawater to hydrous fluids and modify the following sentence accordingly. I agree that serpentinization is usually thought as a seawater-rock interaction. However, serpentinization can occur with any hydrous fluid and in settings different than the seafloor. On the context of this manuscript the serpentine being brought up is a result of water/rock interactions in the mantle wedge and the fluid is likely different in composition than seawater.

Line 104 Authors mention a “very good correlation” between the two isotope systems. I think this is correct for the pore fluid data with seawater fingerprint. However, their figure is shown in a way that appears to indicate that the correlation continues through their serpentine mud data. This is an artifact of the break in the y-axis (figure from table S1 data below). They also left out a single data point for mantle derived fluid that I think should be added to the figure.

Authors also need to clarify how they derived the serpentine mud data point. Their Table S1 does not have pore fluid data for $d_{30}\text{Si}$ 1498B and Table S2 does not have serpentine mud data for $d_{11}\text{B}$. My

guess is that the authors are pairing “mantle-like” fluid $\delta^{11}\text{B}$ from Fantangisña 1498B with $\delta^{30}\text{Si}$ measured in the serpentine muds from 1498B but it is not clear from the figure or the text, if this is the case it needs to be stated in the text or the figure caption as the authors are combining different measurements to get to that data point ($\delta^{30}\text{Si}$ measured in the serpentine muds and $\delta^{11}\text{B}$ measured in pore fluids with mantle derived signature) while leaving out their only data point with both measurements for mantle-derived signature.

Line 138 Typo “serpentinized” missing an i.

Line 160 I think that there is a typo “slap” should be slab.

Line 161-163 This sentence needs some rewriting to improve clarity on what is undersaturated with respect to serpentine.

Line 174 Minor detail that needs to be corrected in section 1.2.1 line 137 d30Si for serpentine is show as 0.3 ± 0.2 here it is shown are 0.33. Authors should be consistent with the value throughout the manuscript.

Response Letter

Kiel, 25.08.2020

Manuscript No.: NCOMMS-19-35155

Title: Serpentine alteration and the impact on the marine Si cycle

Dear Kyle Frischkorn,

Please find enclosed our revised manuscript '*Serpentine alteration and the impact on the marine Si cycle*'. We are very pleased that our revised manuscript was appreciated by the reviewers and that their responses were mainly positive.

We have incorporated the suggestions from reviewer #1 and #3 regarding Fig. 2 by 1) including the mantle-derived fluid data point, 2) calculating and including a mixing curve between seawater and the mantle-derived fluid (see also supplement for calculations), and 3) removing the break on the y-axis. Further, the model code was uploaded in the journal online system and the minor comments by the reviewers were assessed and corrected in the manuscript.

We hope that you agree that the revisions have improved the manuscript and that you will consider the revised version for publication.

With best regards on-behalf of all co-authors
Sonja Geilert

REVIEWER COMMENTS

Reviewer #1 (Remarks to the Author):

Re-review of manuscript NCOMMS-19-35155 submitted to Nature Communications by Sonja Geilert and colleagues: Serpentine alteration and the impact on the marine Si cycle

With apologies that this review is slightly overdue.

Geilert and colleagues present a manuscript that details silicon, boron, and strontium isotope geochemistry ($\delta^{30}\text{Si}$, $\delta^{11}\text{B}$, $87\text{Sr}/86\text{Sr}$) of fluids and muds from seamounts experiencing serpentinization and subsequent serpentine alteration. They argue for no significant Si isotope fractionation during serpentinization reactions, but substantial fractionation during the subsequent dissolution of serpentine and (re)precipitation of authigenic mineral phases (that they represent by talc in their model). This creates fluids with $\delta^{30}\text{Si}$ locally $>5\%$, the highest fluids measured to date, and a potentially important source in the ocean Si budget. In this revised version, the manuscript benefits from a more quantitative model and clarification of several points. I think the central conclusion – that high $\delta^{30}\text{Si}$ fluids are generated by the alteration (and reprecipitation) of serpentine minerals/muds – is robust, although the complexities of natural systems will always hinder any attempts to numerically model the processes. While I have some quibbles about the RTM, I think the novelty of the data and the general interpretative framework are sufficient to warrant publication.

In my original review of the manuscript, I raised several (overlapping) points that are now largely dealt with in the revised version. The largest change is the incorporation of a reactive transport model, and this helps to address many of the issues. Specifically:

- the descriptive nature of the results: the new model goes some way to addressing this. As and aside, I would also encourage the authors to make their model code accessible. Perhaps I miss it in the online portal, but simply displaying an advection-diffusion-reaction equation (L671/SL 17) is not the same as making the underlying scripts available.

The MATHEMATICA model code was uploaded in the online portal of the journal.

- the generalizability of the interpretations to other regions of the ocean: the authors have made a good attempt at justifying the extension of their results to elsewhere (e.g. revised ms section 1.3), while also emphasising that this would need to be demonstrated in other field studies. I find this reasonably convincing.

- the role of temperature as a potential control on fractionation: I had misinterpreted the potential range of temperature experienced by their fluids today (i.e. only with a few deg C, L78 in revised ms). However, the response does not account for the thermal history of the circulating seawater, but I interpret from e.g. Fig 4d that the penetration of seawater into the seamounts is not deep – and therefore not at elevated T, as otherwise it would have distinct Sr isotope composition etc.? Perhaps mention this around L78 for the sake of clarity, if so.

A note was added in lines 83-86 regarding the penetration depth of seawater.

- issues with the numerical model previously used (e.g. assumptions associated with Rayleigh distillation model, etc.). These are largely resolved with the RTM model now employed, though other issues now appear. For example, model sensitivity to the prescribed parameters (Tables S3,4,5) is not investigated. Perhaps this is not necessary: my understanding is that the true value of model lies not in its ability to pinpoint one precise set of parameters, but rather to show that the dissolution of serpentine and subsequent re-precipitation of secondary phases (whether talc, or something else) can produce fluid $\delta^{30}\text{Si}$ in the same range as the observations. One thing that I missed is a quantification of the fraction of Si being re-sequestered into the talc. Presumably there is a trade-off between fluid $\delta^{30}\text{Si}$ and fluid [Si] - as $\delta^{30}\text{Si}$ increases, [Si] decreases. Does this have implications for the importance of the flux at a regional- to global-scale?

We have investigated sensitivity tests for the Si isotope fractionation with different $\Delta^{30}\text{Si}$ shown in Fig. 6c, which is one of the least constrained variables in the model. The reviewer is correct, that further sensitivity tests were not carried out. The model was set up to show general pathways of processes during serpentine alteration. We are not able to derive reliable values for transport velocities, mixing rates, and reaction rates because our simple 1-D steady state model does not resolve the 3-dimensional spatial structure of the two different seamounts and the temporal variability of transport processes. Given these model limitations -that are acknowledged in the main manuscript- we decided to not conduct further sensitivity tests. The depth-integrated rates of serpentine dissolution and authigenic mineral precipitation calculated in the model nearly balance each other out (773 versus 768 $\mu\text{M Si cm}^{-2} \text{ yr}^{-1}$, respectively) so that about 99% Si is removed from pore fluids during authigenic mineral formation (now stated in lines 199-200 in the revised manuscript). This would result in a flux of 5 $\mu\text{M Si cm}^{-2} \text{ yr}^{-1}$. However, given that the model cannot sufficiently resolve the 3D nature of the system (as also stated in the manuscript) we refrain from transferring the flux results to a regional- and global-scale.

The issues raised by other reviewers included the utility of $\delta^{11}\text{B}$, and the ability of extrapolating insights or fluxes from serpentine alteration reactions (as done here) to the broader suite of basalt alteration reactions. I think the authors make a reasonable case in the revised version as to the benefits of a dual-isotope approach, and the arguments based on secondary phase mineralogy are helpful in assessing the extension of the conclusions to other regions of the ocean. However, it seems to me that $\delta^{11}\text{B}$ is not being used as a well-understood tracer to 'calibrate' the $\delta^{30}\text{Si}$ data, as the text/rebuttal seems to imply, but rather as a system that is thought to largely be sensitive to the same processes (i.e. secondary phase formation), so that correlation between the two supports the interpretations, and aids in their model parameterisation. I think this is OK, but if this is not the case

then the text should be clarified to highlight where some knowledge of the system is inferred on the basis of $\delta^{11}\text{B}$ and/or [B] alone, and then the silicon isotope interpretation made on the basis of this knowledge.

We agree with the reviewers comment that both isotope systems show a similar response to fluid-rock reactions and both isotope systems were used to identify the involved processes. We followed the suggestion of the reviewer and did not change the phrasing in the main text.

Some minor comments on the revised version of the manuscript:

L151: "Motivated".

The word was corrected.

L162: "the rapid dissolution of serpentinite, which is undersaturated with respect to serpentinite" – something is off here, please rephrase.

The sentence was revised.

Figure 7: Not clear what the line is. Are the north Pacific data points not the most relevant here?

The mixing line and data are now explained in more detail in the figure caption. The line represents the mixing line between the water masses from the Cascadia Basin and from the Subantarctic Zones. Low Si isotope values in the N and NW Pacific are still not understood.

Are all figures necessary in the main text? It seems like 3 and 4 could potentially be combined, while 5 might be better suited in the supplement. On figure 2, is it possible to display a) mixing hyperbolas between seawater and mantle fluid, and b) model predicted vectors of fluid evolution in $\delta^{30}\text{Si}$ - $\delta^{11}\text{B}$ space? This would allow the reader to assess the extent to which the general assumptions implicit in the RTM can replicate the range of data observed.

Given that the current number of figures is still below the maximum number allowed by the journal, we have decided to leave all figures in the main text.

Regarding Fig. 2, we have included a mixing curve between seawater and the deep mantle fluid. We decided not to include the model data in Fig. 2, because we did not want to shift the focus of Fig. 2 from the measured data to the model data. Further, the current logic and structure of the manuscript would be affected and the incorporation of the model data would require re-writing and re-organization of the manuscript.

Reviewer #2 (Remarks to the Author):

Dear editor,

this revised manuscript has addressed most of the comments from the reviewers. I am happy with their revision. I would like to suggest acceptance if the authors can address a few more things.

1. A few words need further clarification.

line 139: change Si fractionation to Si isotope fractionation.

The word 'isotope' was included in the sentence.

line 149: what is "isotope values"?

The expression was changed to 'isotope value of both elements'.

line 241: change "More data is needed" to "More data are needed".

The word was changed to plural.

2. line 57-58: "B is an incompatible element". "incompatible element" is not referred to the partitioning behaviour between fluid and solid, but melt and solid.

We have rephrased the sentence in lines 61-63: '...B isotopes have been used to study mineral reactions as B is a fluid mobile element and high amounts of B can be incorporated in the serpentine crystal lattice, when B concentrations in the fluids are high...'

3. discussion about altered oceanic crust. line 275-277 and line 281-283. Yu et al. (2018 Chemical Geology) published Si isotope data for altered oceanic crust. the observation should be connected with the discussion here if the authors do want to predict the Si isotope fractionation of altered oceanic crust.

The reference Yu et al. (2018) was added and we discuss now the Si isotope value of the altered oceanic crust. As the authors state, the similarity in $\delta^{30}\text{Si}$ between unaltered and altered oceanic crust might relate to the low degree of alteration (<10%), which is too small to influence bulk $\delta^{30}\text{Si}$ values (lines 286-290 in the revised manuscript).

all the best,

-Fang Huang

Reviewer #3 (Remarks to the Author):

Review of Serpentine alteration and the impact on the marine Si cycle

Authors Sonja Geilert, Patricia Grasse, Klaus Wallmann, Volker Liebetrau, Catriona D. Menzies

General comments

All my comments from my previous review have been addressed in a satisfactory way. They provide an updated manuscript. It retains their Si and B isotope data and provide a novel approach to understand the role of serpentinites and its alteration in the Si global cycle. They added a new section using a one-dimensional transport-reaction model to explain their data and estimate fractionation factors during serpentinite formation and alteration. Dealing with open systems is always problematic and the authors have done a good work in addressing the problematics and come with a model to interpret their data that is solid. Overall their model requires significant serpentine dissolution and loss of some Si from the serpentine muds to seawater. This is shown using pore fluid $\delta^{30}\text{Si}$ and I think is a solid result. However, before I recommend this paper for publication it requires some minor changes in the text as well as some important clarifications with data treatment and figure 2.

Specific comments

Line 30 I suggest that the authors change seawater to hydrous fluids and modify the following sentence accordingly. I agree that serpentinization is usually thought as a seawater-rock interaction. However, serpentinization can occur with any hydrous fluid and in settings different than the seafloor. On the context of this manuscript the serpentine being brought up is a result of water/rock interactions in the mantle wedge and the fluid is likely different in composition than seawater.

The word seawater was replaced by hydrous fluids accordingly.

Line 104 Authors mention a "very good correlation" between the two isotope systems. I think this is correct for the pore fluid data with seawater fingerprint. However, their figure is shown in a way that

appears to indicate that the correlation continues through their serpentine mud data. This is an artifact of the break in the y-axis (figure from table S1 data below). They also left out a single data point for mantle derived fluid that I think should be added to the figure.

We agree that the figure implied a wrong impression of the data and we changed it accordingly (removal of the y-axis break). Additionally, we added the data point from the mantle-derived fluid and emphasized in the caption that the $\delta^{30}\text{Si}$ value may be affected by precipitation during fluid ascent.

Authors also need to clarify how they derived the serpentine mud data point. Their Table S1 does not have pore fluid data for d30Si 1498B and Table S2 does not have serpentine mud data for d11B. My guess is that the authors are pairing “mantle-like” fluid d11B from Fantangisña 1498B with d30Si measured in the serpentine muds from 1498B but it is not clear from the figure or the text, if this is the case it needs to be stated in the text or the figure caption as the authors are combining different measurements to get to that data point (d30Si measured in the serpentine muds and d11B measured in pore fluids with mantle derived signature) while leaving out their only data point with both measurements for mantle-derived signature.

We added an explanation for the serpentine mud data point in the caption and included the data point from the mantle-derived fluid.

Line 138 Typo “serpentinized” missing an i.

The typo was corrected.

Line 160 I think that there is a typo “slap” should be slab.

The typo was corrected.

Line 161-163 This sentence needs some rewriting to improve clarity on what is undersaturated with respect to serpentine.

The sentence was revised.

Line 174 Minor detail that needs to be corrected in section 1.2.1 line 137 d30Si for serpentine is show as 0.3 ± 0.2 here it is shown are 0.33. Authors should be consistent with the value throughout the manuscript.

The second digit was removed in line 181 and checked throughout the text, based on the size of the standard deviation (2SD = 0.2‰).

Serpentine alteration and the impact on the marine Si cycle

Sonja Geilert^{1*}, Patricia Grasse^{1,2}, Klaus Wallmann¹, Volker Liebetrau¹, Catriona D. Menzies³

¹GEOMAR Helmholtz Centre for Ocean Research Kiel, Wischhofstr. 1-3, 24148 Kiel, Germany

²German Centre for Integrative Biodiversity Research (iDiv) Halle-Jena-Leipzig, Deutscher Platz 5e, 04103 Leipzig, Germany

³Department of Earth Sciences, Durham University, Science Laboratories, South Road, Durham, UK

[*corresponding author; sgeilert@geomar.de](mailto:sgeilert@geomar.de)

Keywords: Mariana seamounts, serpentine alteration, Si isotopes, B isotopes, authigenic clay formation, marine Si cycle

Abstract

[revised manuscript text omitted]
^{11}\text{B}$ ($+16.1$ to $+43.5\text{‰}$), which encompasses $\delta^{11}\text{B}$ of
104 seawater (seawater B concentration: $432.6 \mu\text{M}^{29}$; $\delta^{11}\text{B}_{\text{SW}}$: $+39.6\text{‰}^{30}$) (Fig. 1c).
105 The combination of pore fluid $\delta^{11}\text{B}$ and $\delta^{30}\text{Si}$ values shows a very good correlation of these
106 two isotope systems (Fig. 2). For both systems, the isotope values are below and above the
107 seawater signature, indicating dissolution of presumably serpentine and precipitation of
108 authigenic mineral phases, respectively. In order to decipher the processes controlling
109 mineral dissolution and precipitation, fluid sources and compositional changes in relation to
110 depth need to be examined.

111

112 *1.1 Origin of pore fluids*

113

114 Two major fluid sources can be distinguished based on magnesium (Mg^{2+}), strontium (Sr),
115 and chloride (Cl^-) concentrations, radiogenic Sr isotopes, and pH (Fig. 3, 4a-c).

116 Most of the pore fluids show seawater-like signatures in the upper ~ 30 m (surface layer),
117 except samples from Fantangisña sites 1498A and B (Fig. 3, 4a-c). Samples from site 1498B
118 overlap with the local fluid mantle endmember and can thus be identified as mantle fluids,
119 showing depleted Mg^{2+} concentrations, high pH values (~ 11), decreasing Cl^- concentrations,
120 and increasing Sr concentrations, characteristic trends reported for deep mantle fluids from
121 the Mariana forearc region^{26,31}. Also, a $^{87}\text{Sr}/^{86}\text{Sr}$ ratio similar to deep-sourced mantle fluids is
122 measured ~~in these fluids~~ ($^{87}\text{Sr}/^{86}\text{Sr}$ of 0.70535: ODP Site 1200 at South Chamorro
123 Seamount²⁶) (see also lower red circle in Fig. 1b). Therefore, we interpret the related Si and
124 B isotope values to originate from the deep mantle affected by pervasive serpentinization.
125 Sample 1498A 15-R-1 shows a high Sr concentration ($189 \mu\text{M}$) and a more radiogenic
126 $^{87}\text{Sr}/^{86}\text{Sr}$ signature (0.70763; Fig. 4) compared to the mantle-derived fluids and likely results
127 from mixing between the two endmembers (seawater and deep-mantle fluid) discussed
128 above.

129

1.2 Identification of fractionation processes

1.2.1 Drivers of deep mantle-derived $\delta^{30}\text{Si}$ and $\delta^{11}\text{B}$ values

During serpentinization, pore fluid pH can increase rapidly to alkaline values (pH ~9 - 12²⁸), so that dissolved B is tetrahedrally coordinated. This is also the species preferentially bound in the serpentine mineral structure, and so limited fractionation between mineral and water is expected taking also the high formation temperatures into account (average T of 200°C)^{20,32}. The mantle-derived pore fluids of site 1498B have the lowest $\delta^{11}\text{B}$ values of about +16‰ and the lowest B concentrations of the investigated fluids (between 84 and 111 μM ; Table S1). These low $\delta^{11}\text{B}$ values overlap within error with serpentinized peridotite clasts and serpentinite matrix identified by Benton et al.¹⁸ with $\delta^{11}\text{B}$ values of about +14‰ for Conical seamount in the Mariana forearc. Consequently, we conclude that no significant B isotope fractionation occurs during early pervasive high pH serpentinization reactions. In contrast, Si isotopes vary between potentially mantle-derived fluids ($\delta^{30}\text{Si} = +1.6\text{‰}$) and serpentinite muds (on average $\delta^{30}\text{Si} = -0.3 \pm 0.2\text{‰}$; 1SD; Table S2). However, by comparing pristine mantle rocks ($\delta^{30}\text{Si} = -0.29 \pm 0.08\text{‰}$)³³ with the serpentinized muds investigated in this study, we show no, or only minor, Si isotope fractionation occurs during the transformation of olivine/ pyroxene to serpentine. This also confirms the isochemical nature of pervasive serpentinization³. The pore fluid $\delta^{30}\text{Si}$ value is likely affected during ascent of the serpentinite muds and accompanied cooling, which induces Si precipitation and fractionation of Si isotopes.

1.2.2 Processes of serpentine alteration

The investigation of Si and B isotopes revealed a similar fractionation response during serpentine alteration and shows the potential of coupled Si and B isotope data to trace serpentine alteration reactions (Fig. 2). Interestingly, despite generally increasing Si and B concentrations, the isotope values of both elements increase as well. This observation seems counterintuitive, in that concentration increases generally relate to mineral dissolution processes, which are not associated with significant isotopic fractionation³⁴. Motivated by this observation, a one-dimensional transport-reaction model (see Methods and

162 supplementary materials for model details) was set up to constrain— for the first time— the
 163 seawater entrainment rate, the precipitation and dissolution rates, and the isotopic
 fractionation during authigenic mineral formation. This model is a simplification as it does
 not capture the additional changes which are likely induced by the 3-D geometry of the
 seamounts, and the time- and space-dependency of mixing rates and fluid flow velocities.
 The depth profiles indicate a seamount surface layer with a thickness of about 30 m where
 bottom water is entrained into the muds by shallow seawater circulation and an underlying
 layer that is affected by the ascent of deep fluids originating from the subducted ~~slap-slab~~
 (Fig. 3, 4). The strong increase in Si and B concentrations in the surface layer is induced by
 rapid dissolution of serpentine, which is undersaturated ~~with respect to serpentine~~ due to
 the entrainment of pH-neutral seawater (Fig. 5). The persistent mixing in the surface layer
 with seawater supports high dissolution rates ($\sim 1.2 \mu\text{mol cm}^{-3} \text{ yr}^{-1}$) whereas the rates are low
 in the deeper layers ($< 10^{-5} \mu\text{mol cm}^{-3} \text{ yr}^{-1}$) that are not affected by mixing with ambient
 seawater (Fig. 5). Serpentine, which formed in the mantle wedge, becomes unstable after
 deposition on the seafloor either as serpentized clasts or as serpentinite mud and begins
 to dissolve during interaction with seawater via hydrolysis³⁵, following:

thereby releasing Si and B into the pore fluid (see also upper red circle in Fig. 1b; Fig. 6a,b).
 The $\delta^{30}\text{Si}$ and $\delta^{11}\text{B}$ values increase in the about upper 30 m of the seamounts even though
 isotopically depleted Si ($\delta^{30}\text{Si} = -0.33\%$; Table S2) and B ($\delta^{11}\text{B} = +16\%$; see section 1.2.1) are
 released by the dissolving serpentine. This observation can only be explained by formation
 of authigenic minerals removing isotopically depleted Si and B from the pore fluids (Fig. 2,
 46). The formation of authigenic minerals was simulated by allowing talc ($Mg_3Si_4O_{10}(OH)_2$)
 precipitation in the model. Talc was chosen because thermodynamic equilibrium calculations
 conducted with PHREEQC³⁶ show that the surface layer of the seamounts is strongly
 oversaturated with respect to this mineral (Fig. 5). Other Al-containing silicate phases may
 form as well; however, we were not able to simulate the formation of these phases due to
 the lack of dissolved Al data. The modeling indicates that a large fraction of the Si and B
 released from the dissolving serpentine is removed from solution by authigenic mineral
 precipitation (Fig. 5). We obtained a good fit to the $\delta^{30}\text{Si}$ data applying a Si isotope

fractionation of $\Delta^{30}\text{Si} = -3\text{‰}$, where $\Delta^{30}\text{Si}$ is defined as $\Delta^{30}\text{Si} = \delta^{30}\text{Si}_{\text{mineral}} - \delta^{30}\text{Si}_{\text{pore fluid}}$. Lower
and higher Si isotope fractionation could not reproduce the observed $\delta^{30}\text{Si}$ values of the
sample as tested by model runs using $\Delta^{30}\text{Si}$ of -2‰ and -4‰ (Fig. 6c). This Si isotope
fractionation is the highest abiotic fractionation observed for authigenic mineral formation
so far. In natural systems, only Si precipitation from supersaturated solutions to form
amorphous silica and experimentally investigated Si adsorption to Al shows $\Delta^{30}\text{Si}$ values in
this range or higher (up to -5‰)³⁷⁻³⁹.

| The almost complete removal of dissolved Si by authigenic mineral precipitation (~99% Si
removed from pore fluid) induces a strong $\delta^{30}\text{Si}$ maximum at the base of the surface layer,
which is the highest pore fluid $\delta^{30}\text{Si}$ value that has been observed in natural environments.
There is only a limited data set available on marine pore fluid $\delta^{30}\text{Si}$, mostly originating from
continental margin settings. The $\delta^{30}\text{Si}$ values range between -0.5 to $+2.5\text{‰}$ and are
dominantly controlled by dissolution of biogenic silica and the formation of authigenic clays
with a $\Delta^{30}\text{Si}$ of -2‰ ⁴⁰⁻⁴². The Si isotope fractionation depends on the chemical composition
and crystal structure of the authigenic mineral, the precipitation rate, and temperature, and
successive mineral precipitation^{37,38,43,44}. We can only speculate that a combination of slow
reaction rates associated with low temperatures (authigenic mineral precipitation rate ~ 1.2
$\mu\text{mol Si cm}^{-3} \text{ yr}^{-1}$ at $\sim 4^\circ\text{C}$; Fig. 5) and a Mg-rich authigenic mineral composition drives the Si
isotope fractionation to higher values at the Mariana seamounts compared to continental
margin settings (e.g. Peruvian margin) with faster reaction rates at higher temperatures
(authigenic mineral precipitation rate up to $\sim 27 \mu\text{mol Si cm}^{-3} \text{ yr}^{-1}$ at $\sim 11^\circ\text{C}$; Ehlert et al.⁴⁰) and
| an Al-rich authigenic mineral composition. This assumption needs to be constrained-tested
in further studies, however, similar reactions and associated Si isotope fractionation as
observed for the Mariana seamounts may occur in other settings, where oceanic crust and
serpentinite interact with seawater at low temperatures (see section 1.3).

The isotopic fractionation of B during mineral precipitation is related to two processes: the
pH-dependent fractionation between borate and boric acid in the pore fluid and the
fractionation during borate uptake in the solid phase³². Hence, we calculated the
concentrations of borate and boric acid and their isotopic composition applying the
dissociation constant for boric acid in seawater⁴⁵ and the isotopic equilibrium constant for
borate and boric acid⁴⁶. The model results confirm that dissolved borate has a much lower

$\delta^{11}\text{B}$ than total dissolved boron in the surface layer (Fig. 6d) such that ^{10}B is removed from
solution when borate is preferentially bound in the authigenic silicate phase. The modeling
showed that an additional fractionation between dissolved borate and solid phase borate
has to be applied to match the $\delta^{11}\text{B}$ data. The best fit was attained applying an isotopic
fractionation factor of $\Delta^{11}\text{B}_{\text{mineral-borate}}$ of -20‰ where $\Delta^{11}\text{B}_{\text{mineral-borate}}$ is defined as $\delta^{11}\text{B}_{\text{borate}}$
$\text{authigenic mineral} - \delta^{11}\text{B}_{\text{dissolved borate}}$. This B_{borate} isotope fractionation results in an average $\delta^{11}\text{B}$
value of the authigenic clay of $-2.4 \pm 2.0\%$ (1 SD), supposing an average of the modeled
$\delta^{11}\text{B}_{\text{borate}}$ in the upper 15 m (highest talc precipitation rate) of +17.6‰. This results in a
general B isotope fractionation between authigenic mineral and the pore fluid
of $-43.3 \pm 2.3\%$ (1 SD), with $\Delta^{11}\text{B}_{\text{mineral-pore fluid}} = \delta^{11}\text{B}_{\text{mineral}} - \delta^{11}\text{B}_{\text{pore fluid}}$ (average of the modeled
$\delta^{11}\text{B}_{\text{pore fluid}}$ in the upper 15 m of +40.9‰). This B fractionation fits well to modeled values for
low temperature phyllosilicate formation by Boschi et al.¹⁹ based on data by Liu & Tossel³².

[revised manuscript text omitted]

measurement of altered oceanic crust has an average $\delta^{30}\text{Si}$ value of $-0.32\pm 0.06\%$ ⁵⁶,
overlapping with pristine mantle rocks ($\delta^{30}\text{Si} = -0.29\pm 0.08\%$)³³. However, this value of
altered oceanic crust originates from the east side of the East Pacific Rise, where the
alteration degree is $<10\%$ ⁵⁷ and thus too small to influence bulk rock $\delta^{30}\text{Si}$ values. The impact
on oceanic $\delta^{30}\text{Si}$ values needs to be assessed by constraining the Si fluxes and associated
$\delta^{30}\text{Si}$ values related to serpentine and basalt alteration and authigenic mineral formation.
Oceanic crust alteration may also influence oceanic $\delta^{30}\text{Si}$ values at regional scale, especially
in restricted basins and may potentially explain, at least in part, Si anomalies the Northeast
Pacific Silicic Acid Plume (NPSP). The NPSP has unusually high Si concentrations ($> 150 \mu\text{M Si}$)
and high $\delta^{30}\text{Si}$ signatures of $+1.5\%$, the highest Pacific deep ocean Si isotope value^{58,59} (Fig.
7), between 2000 and 3000 m water depths, originating mostly from the Cascadia Basin
(about $200 \mu\text{M Si}$)⁶⁰. This has not been explained unambiguously to date. Findings of this
study show that low temperature alteration of serpentinites and potentially seafloor basalts
are associated with high Si concentration and $\delta^{30}\text{Si}$ values which are expelled to the ocean.
Such a source is likely to contribute to the unusually high NPSP and related Si isotope values
(see Supplement for details). Low temperature seafloor may contribute significantly to the
$\delta^{30}\text{Si}$ budget of the ocean, considering the large areas of serpentinized oceanic crust and the
related fluid discharge with extremely high Si isotope values.

**Acknowledgements**

We appreciate the support of the master and crew of the JOIDES Resolution during the IODP
Expedition 366 Science Party. We thank Regina Surberg and Ana Kolevica for analytical support.
Further thanks go to Jutta Heinze, Daphne Bartels, and Tyler Goepfert. We are grateful to an
anonymous reviewer, F. Huang, and J.C. de Obeso for their constructive reviews and thoughtful
comments. IODP Germany and ECORD are thanked for financial support as well as IODP grant code:
NERC UK IODP Phase 2 Moratorium Award NE/P020909/1.

**Author contributions**

CM was involved in research cruise IODP366 and carried out the sampling. SG and PG performed the
Si isotope analyses. SG performed the B isotope analyses. VL conducted the Sr isotope
measurements. KW designed the transport-reaction model. SG, PG, CM, and KW interpreted the
isotope data. SG wrote the manuscript with help from PG, CM, VL, and KW.

**Competing interests**

The authors declare no competing interests.

**Data availability**

[revised manuscript text omitted]

**Figures**

**Fig. 1. Location and isotope results of Mariana seamount pore fluids.** (a) Sampling area of the
 two seamounts (Yinazao and Fantangisña) investigated during IODP Expedition 366 and close
 up of the sample locations for both seamounts. (b) Sketch of the Mariana forearc region with
 indicated sampling locations (modified after Fryer et al.²⁸). Blue dotted lines indicate fluid
 flow along faults. Note mantle-derived fluid flow for site 1498B (see section 1.1 and 1.2.1).
 (c) Isotope compositions versus depths (m) are displayed for $\delta^{30}\text{Si}$ and $\delta^{11}\text{B}$. Error bars are
 within symbol size. Brown area denotes depth of uppermost pelagic clays (≤ 5 m depth) and
 green area shows mineralogy dominated by serpentinites and mafic xenoliths.

Fig. 2. Pore fluid $\delta^{11}\text{B}$ versus $\delta^{30}\text{Si}$ values reveal different stages of silicate dissolution and
 precipitation (see text for explanation). The mixing curve is calculated following equation 1
 (supplement) between seawater and a mantle-derived fluid. The grey-shaded area takes the
 varying low Si concentrations of the mantle-derived fluids into account (see Supplement for
 details). Error bars (2SD) not indicated are within symbol size. Note that the $\delta^{30}\text{Si}$ value of
 the mantle-derived fluid is likely affected by Si precipitation during fluid ascent (for details
 see section 1.2.1) and that the data point from the serpentinite mud is derived from direct
 $\delta^{30}\text{Si}$ measurements (Table S2) and inferred for $\delta^{11}\text{B}$ from pore fluids given that no
 fractionation occurs during serpentinization (section 1.2.1).

Fig. 3. Depth-profiles characterizing the layering of the seamount fluid system. a) Mg²⁺
 concentrations in pore fluids, b) pH values, c) sediment porosity. Functions were fitted
 through the data to generate continuous depth profiles for the transport-reaction modeling
 (supplementary information). Mg²⁺ and porosity data are reported in Fryer et al.²⁸.

Fig. 4. Depth profiles of conservative tracers. a) Dissolved Sr, b) $^{87}\text{Sr}/^{86}\text{Sr}$ ratios in pore fluids,
 c) Dissolved Cl^- (taken from Fryer et al.²⁸), d) The non-local mixing coefficient α was derived
 by fitting the model to the data (a-c). Dissolved Cl^- , Sr, and $^{87}\text{Sr}/^{86}\text{Sr}$ ratio in the pore fluid
 samples were employed to constrain the steady-state mixing rate and fluid flow velocity
 assuming that the depth-distribution of these dissolved tracers is governed by transport
 processes rather than precipitation/dissolution reactions. The best fit to these data was
 obtained applying mixing in the upper 30 m with a non-locale mixing coefficient of 5×10^{-4}
 582 yr^{-1} and an upward fluid flow velocity of 0.01 cm yr^{-1} . Thick lines indicate steady-state model
 results (supplementary information).

Fig. 5. Reaction rates applied in the modeling to obtain the model fit shown in Fig. 6. a)
 Saturation state of the pore fluid with respect to serpentine, b) saturation state of the pore
 fluid with respect talc. Saturation states were calculated applying measured pH values and
 590 Mg^{2+} concentrations (Fig. 3) and modeled silica concentrations (Fig. 6). c) Rate of serpentine
 dissolution, d) rate of talc precipitation. Note the different depth scale in (c) and (d).

Fig. 6. Depth profiles of reactive tracers. a) Dissolved Si, b) total dissolved B, dissolved borate
 | (B(OH)₄⁻) and dissolved boric acid (B(OH)₃), c) δ³⁰Si of dissolved ~~silica~~Si. Model data based on
 | a Si isotope fractionation with Δ³⁰Si of -3‰ (best fit) and sensitivity tests assuming -2‰
 | and -4‰ (dotted lines). d) δ¹¹B values of total dissolved ~~boron~~B, dissolved borate and
 | dissolved boric acid following Zeebe & Wolf-Gladrow⁴⁵ and Klochko et al.⁴⁶ (see section 1.2.2
 | and Supplement for details). Error bars within symbol size. Thick or labelled lines indicate
 | steady-state model results (supplementary information).

[revised manuscript text omitted]

$+19.7\pm 0.2\text{‰}$ (2 SD; $n=123$), and ERM-AE122 with $+39.7\pm 0.2\text{‰}$ (2 SD; $n=104$). The samples

were measured 2-times per sequence on two to three days and had reproducibilities
between 0.1 and 0.3 ‰ (2SD; Table S1).

***Strontium sample preparation and isotope analyses by TIMS***

The Strontium isotope ratios were measured by Thermal Ionization Mass Spectrometry
(TIMS, Triton, ThermoFisher Scientific). The samples were chemically separated via cation
exchange chromatography using the SrSpec resin (Eichrom). The isotope ratios were
normalized to NIST SRM 987, which has a $^{87}\text{Sr}/^{86}\text{Sr}$ ratio of 0.710248¹⁵ and the
measurements yielded a precision of ± 0.000015 (2 SD, n = 12).

**Numerical transport-reaction modeling**

A transport-reaction model was set up to simulate the processes occurring in the surface
zone of serpentinite seamounts. The 1-D pore fluid model considers molecular diffusion,
upward fluid flow, bottom water intrusion in surface sediments, serpentine dissolution, and
talc precipitation. The turnover of dissolved species in the pore fluid is simulated applying
the following mass balance equation:

$$\Phi \cdot \frac{\partial C}{\partial t} = \frac{\partial}{\partial x} \left(\Phi \cdot \left(D_s \cdot \frac{\partial C}{\partial x} + v \cdot C \right) \right) + \Phi \cdot \alpha \cdot (C_{BW} - C) + \Phi \cdot R$$

with Φ : porosity, C : concentration of dissolved species in the pore fluid ($\mu\text{mol cm}^{-3}$), t : time
(yr), x : sediment depth (cm), D_s : molecular diffusion coefficient of dissolved species in
sediment pore fluid ($\text{cm}^2 \text{ yr}^{-1}$), v : upward fluid flow velocity of pore fluid (cm yr^{-1}), α : mixing
coefficient (yr^{-1}), R : turnover rates of dissolved species ($\mu\text{mol cm}^{-3} \text{ yr}^{-1}$). The model was set
up for the following dissolved species: Si, ^{30}Si , B, ^{11}B , Sr, ^{87}Sr , Cl. The isotopic compositions of
the pore fluids ($\delta^{30}\text{Si}$, $\delta^{11}\text{B}$, $^{87}\text{Sr}/^{86}\text{Sr}$) are calculated from the corresponding mole fractions
($^{30}\text{Si}/\text{Si}$, $^{11}\text{B}/\text{B}$, $^{87}\text{Sr}/\text{Sr}$) applying previously published approaches^{7, 16} and the boundary
conditions, equations and parameter values given in the supplementary materials.

**Method References**

- 1. Fryer, P., Wheat, C. G., Williams, T. & Expedition 366 Scientists, T. Mariana Convergent
Margin and South Chamorro Seamount. *Proc. Int. Ocean Discov. Program, 366 Coll.*
*Station. TX (International Ocean Discov. Program) (2018).*
- 2. Gieskes, J. M., Gamo, T. & Brumsack, H. Chemical methods for interstitial water
analysis aboard Joides Resolution. *Ocean Drill. Prog. Tech. Note 15. Texas A&M Univ.*
*Coll. Stn. (1991).*
- 3. van den Boorn, S. H. J. M., Vroon, P.Z., van Belle, C.C., van der Wagt, B., Schwieters, J.,
van Bergen, M. Determination of silicon isotope ratios in silicate materials by high-
resolution MC-ICP-MS using a sodium hydroxide sample digestion method. *J. Anal. At.*
*Spectrom. 21*, 734 (2006).
- 4. Georg, R. B., Reynolds, B. C., Frank, M. & Halliday, A. N. New sample preparation
techniques for the determination of Si isotopic compositions using MC-ICPMS. *Chem.*
*Geol. 235*, 95–104 (2006).
- 5. van den Boorn, S. H. J. M., Vroon, P. Z. & van Bergen, M. J. Sulfur-induced offsets in
MC-ICP-MS silicon-isotope measurements. *J. Anal. At. Spectrom. 24*, 1111 (2009).
- 6. Hughes, H. J., Delvigne, C., Korntheuer, M., de Jong, J., André, L., Cardinal, D.
Controlling the mass bias introduced by anionic and organic matrices in silicon
isotopic measurements by MC-ICP-MS. *J. Anal. At. Spectrom. 26*, 1892 (2011).
- 7. Ehlert, C., Doering, K., Wallmann, K., Scholz, F., Sommer, S., Grasse, P., Geilert, S.,
Frank, M. Stable silicon isotope signatures of marine pore waters – Biogenic opal
dissolution versus authigenic clay mineral formation. *Geochim. Cosmochim. Acta 191*,
102–117 (2016).
- 8. Cardinal, D., Alleman, L. Y., de Jong, J., Ziegler, K. & André, L. Isotopic composition of
silicon measured by multicollector plasma source mass spectrometry in dry plasma
mode. *J. Anal. At. Spectrom. 18*, 213–218 (2003).
- 9. Oelze, M., Schuessler, J. A. & von Blanckenburg, F. Mass bias stabilization by Mg
doping for Si stable isotope analysis by MC-ICP-MS. *J. Anal. At. Spectrom. 31*, 2094–
2100 (2016).
- 10. Albarède, F., Telouk, P., Blichert-Toft, J., Boyet, M., Agranier, A., Nelson, B. Precise
and accurate isotopic measurements using multiple-collector ICPMS. *Geochim.*
*Cosmochim. Acta 68*, 2725–2744 (2004).
- 11. Grasse, P., Brzezinski, M., Cardinal, D., de Souza, G. F., Andersson, P., Closset, I., Cao,
Z., Dai, M., Ehlert, C., Estrade, N., Francois, R., Frank, M., Jiang, G., Jones, J.J.,
Kooijman, E., Liu, Q., Lu, D., Pahnke, K., Ponzevera, E., Schmitt, M., Sun, X., Sutton,
794 J.N., Thil, F., Weis, D., Wetzels, F., Zhang, A., Zhang, J., Zhang, Z. GEOTRACES inter-
795 calibration of the stable silicon isotope composition of dissolved silicic acid in
seawater. *J. Anal. At. Spectrom. 32*, 562–578 (2017).
- 12. Gaillardet, J., Lemarchand, D., Göpel, C. & Manhès, G. Evaporation and sublimation of
boric acid: Application for boron purification from organic rich solutions. *Geostand.*
*Newsl. 25*, 67–75 (2001).
- 13. Jurikova, H., Liebetrau, V., Gutjahr, M., Rollion-Bard, C., Hu, M.Y., Krause, S., Henkel,
D., Hiebenthal, C., Schmidt, M., Laudien, J., Eisenhauer, A. Boron isotope systematics
of cultured brachiopods : Response to acidification , vital effects and implications for
palaeo-pH reconstruction. *Geochemica Cosmochimica Acta 248*, 370–386 (2019).
- 14. Vogl, J. & Rosner, M. Production and Certification of a Unique Set of Isotope and Delta
Reference Materials for Boron Isotope Determination in Geochemical, Environmental

- and Industrial Materials. *Geostand. Geoanalytical Res.* **36**, 161–175 (2012).
- 15. Howarth, R. J. & McArthur, J. M. Strontium isotope stratigraphy. in *A Geological Time*
*Scale, with Look-up Table Version 4* (eds. Gradstein, F. M. & Ogg, J. G.) 96–105
(Cambridge University Press, Cambridge, U.K pp. ., 2004).
- 16. Geilert, S., Grasse, P., Doering, K., Wallmann, K., Ehlert, C., Scholz, F., Frank, M.,
Schmidt, M., Hensen, C. Impact of ambient conditions on the Si isotope fractionation
in marine pore fluids during early diagenesis. *Biogeosciences* **17**, 1745–1763 (2020).

REVIEWERS' COMMENTS

Reviewer #3 (Remarks to the Author):

All my comments from my previous review have been addressed in a satisfactory way. I recommend this revision for publication in Nature Communications. The authors have a novel dataset and a good model to interpret their data. Their interpretations provide good insight in the formation and nature of serpentine mud volcanoes and its potential to alter Si signatures in the ocean and suggest that similar processes might be wide spread in basaltic crust.